# Autoencoders that don't overfit towards the Identity

**Harald Steck**
Netflix
Los Gatos, California
hsteck@netflix.com

## Abstract

Autoencoders (AE) aim to reproduce the output from the input. They may hence tend to overfit towards learning the identity-function between the input and output, i.e., they may predict each feature in the output from *itself* in the input. This is not useful, however, when AEs are used for prediction tasks in the presence of noise in the data. It may seem intuitively evident that this kind of overfitting is prevented by training a denoising AE [36], as the dropped-out features have to be predicted from the *other* features. In this paper, we consider linear autoencoders, as they facilitate analytic solutions, and first show that denoising / dropout actually prevents the overfitting towards the identity-function only to the degree that it is penalized by the induced L2-norm regularization. In the main theorem of this paper, we show that the *emphasized* denoising AE [37] is indeed capable of completely eliminating the overfitting towards the identity-function. Our derivations reveal several new insights, including the closed-form solution of the full-rank model, as well as a new (near-)orthogonality constraint in the low-rank model. While this constraint is conceptually very different from the regularizers recently proposed in [11, 42, 14], their resulting effects on the learned embeddings are empirically similar. Our experiments on three well-known data-sets corroborate the various theoretical insights derived in this paper.

## 1 Introduction and Motivation

Autoencoders (AE) have been successful in various unsupervised problems, including machine translation (e.g. [34]), computer vision (e.g., [30]) and recommender systems (e.g., [41]). In machine learning applications, their prediction accuracy on noisy test-data is often more important than the learned encodings/representations. The latter are often simply a means for achieving high prediction accuracy. Given a data-point as input, an AE aims to reconstruct the feature-values of this data-point in its output-layer. Obviously, the trivial yet futile solution is to learn literally the identity-function between the input and output-layer (given sufficiently large model-capacity), i.e., to predict each feature $i$ in the output-layer from the *same* feature $i$ in the input-layer. As to achieve high prediction accuracy on noisy data, however, the AE should ideally learn all the relevant dependences/interactions among the various features, i.e., feature $i$ in the output-layer should be predicted by taking into account all *other* features $j \neq i$ it depends on in the input-layer. Intuitively speaking, when the learned AE makes predictions for a feature $i$ in the output-layer by relying 'too much' on the *same* feature $i$ in the input-layer (i.e., identity function), and 'not enough' on the *other* features $j \neq i$ it depends on, we call this *overfitting towards the identity function* in this paper.[1] Even if the model-capacity is limited, the training may still tend to overfit towards the identity-function, as we will show in this paper.

Applying *denoising* during training was found to be a very effective regularizer [36]. While there are different kinds of noise for corrupting the input (e.g., see [28]), in this paper we focus on denoising by random dropout [18], the most common approach. In recent years, there has been extensive work on understanding denoising and dropout based on different perspectives, like preventing co-adaptation [18, 32], ensemble-averaging [4, 2], marginalized analysis and $L_2$-norm regularization [10, 40, 28, 39, 16, 22, 20, 9] and others [36, 35, 25, 24].

In denoising, when a feature is randomly dropped out from a data-point in the input-layer, then the (uncorrupted) value of this feature in the output-layer has to be predicted based on the *other* features of that data-point. It hence seems intuitively evident that this may prevent the AE from overfitting towards the identity function. The fact that this intuition turns out to be only partially correct, as we will show, motivated this work.

In this paper, we consider the simplified case of a *linear* autoencoder (LAE) trained with least squares, as this facilitates analytic insights. Note that linear models have been used before to better understand deep nonlinear models, e.g., [31, 1, 29, 23]. After introducing notation in Section 2, we first outline the limitations of denoising by dropout (Section 3): given that it is asymptotically equivalent to $L_2$-norm regularization in LAE (see also [7, 10, 40, 28, 39, 16, 20, 9]), we show that overfitting towards the identity is only prevented to the degree that it is penalized by $L_2$-norm regularization.

In Section 4, we show that the so-called *emphasized* denoising AE [37] is indeed able to *completely* prevent overfitting towards the identity-function. The theoretical analysis of the emphasized denoising LAE (EDLAE) is the central result of this paper, from which several new insights can easily be derived: the closed-form solution for the full-rank model, as well as a novel *(near-)orthogonality-constraint* in the low-rank EDLAE. This constraint encourages the learned latent embeddings to become more 'spread out' in the latent space, a similar effect as recently obtained with conceptually very different approaches: Parseval networks [11] to improve robustness to adversarial examples, as well as the spread-out [42] and GLaS [14] regularizers for extreme classification problems. The experiments on three well-known data-sets in Section 6 empirically corroborate the various insights derived in this paper.

## 2 Model Definition and Notation

In this section, we introduce notation and define the training objective of EDLAE. We assume that the training data are given in terms of a matrix $\mathbf{X} \in \mathbb{R}^{r \times m}$ with $r$ data points (rows) and $m$ features (columns). As to facilitate an analytic solution, we focus on *linear* autoencoders (LAE) trained by least squares. Also in [31, 1, 29, 23], linear models were studied with the goal to better understand deep nonlinear models. Let the LAE be represented by a matrix $\mathbf{B} \in \mathbb{R}^{m \times m}$. If $\mathbf{B}$ is of low rank, say of rank $k$, it may be written as $\mathbf{B} = \mathbf{U}\mathbf{V}^\top$, where the rank-$k$ matrices $\mathbf{U}, \mathbf{V} \in \mathbb{R}^{m \times k}$ correspond to the encoder and decoder, respectively.

Denoising by randomly dropping out feature-values in the input-layer of the LAE, may be formalized as follows: assuming that the training is comprised of $n$ epochs of gradient descent, we vertically stack $n$ times the given (finite) training matrix $\mathbf{X}$ as to obtain the (uncorrupted) target-matrix $\mathbf{X}^{(n)} = [\mathbf{X}^\top ... \mathbf{X}^\top]^\top$. Correspondingly, we generate the corrupted input-matrix $\mathbf{Z}^{(n)}$ by applying dropout to each entry (independently of the other entries): $\mathbf{Z}^{(n)}_{u,i} = \delta_{u,i} \cdot \mathbf{X}^{(n)}_{u,i}$ for all $u, i$, where each $\delta_{u,i} \in \{0, 1\}$ is the realization of a Bernoulli variable with success (i.e., 1) probability $q = 1 - p$, where $p$ is the dropout probability. Given that $\mathbf{X}$ has $r$ rows, note that $\mathbf{X}^{(n)}$ and $\mathbf{Z}^{(n)}$ each have $n \cdot r$ rows. We can now write down the training objective, using the squared error, as it facilitates analytic solutions. We consider the asymptotic limit where the number of training epochs $n \to \infty$, i.e., the stochastic dropout-learning has converged:

$$\hat{\mathbf{B}} = \underset{B}{\operatorname{argmin}} \lim_{n \to \infty} \frac{1}{n} \left\| \mathbf{A}^{(n)1/2} \odot \left( \mathbf{X}^{(n)} - \mathbf{Z}^{(n)} \cdot \mathbf{B} \right) \right\|_F^2, \tag{1}$$

where $|| \cdot ||_F$ denotes the Frobenius norm. Following the definition of EDLAE [37], we introduce the weighting matrix $\mathbf{A}^{(n)}$, where $\mathbf{A}^{(n)}_{u,i} = a$ if $\delta_{u,i} = 0$ (where $\delta_{u,i}$ is the same realization as above), and $\mathbf{A}^{(n)}_{u,i} = b$ otherwise, with $a \geq b$. In other words, compared to the default weight $b$, the error in predicting feature $i$ in data point $u$ in the output-layer is up-weighted by $a$ (where $a \geq b$) if this feature was dropped out in the input-layer–and hence has to be predicted based on the *other*

input-features. For this reason, it is apparent that the overfitting towards the identity-function is *completely* prevented for the choice $a > b = 0$ (called *full emphasis* in [37]), as it considers only the errors on those features that were dropped out in the input and hence have to be predicted based on the *other* features. In Eq. 1, $\mathbf{A}^{(n)1/2}$ is the elementwise square-root of $\mathbf{A}^{(n)}$ as it is inside the squared error, and $\odot$ is the elementwise product. Given that dropout is applied only during the training of $\hat{\mathbf{B}}$, the learned parameters have to be re-scaled when making predictions on new data points (without dropout) [18], resulting in the final solution:

$$\hat{\mathbf{B}}^{(\text{EDLAE})} = q \cdot \hat{\mathbf{B}} \tag{2}$$

## 3   Denoising merely induces $L_2$-Norm Regularization

In this section, we first review the (standard) dropout-denoising (i.e., $a = b$) and outline that it is asymptotically equivalent to $L_2$-norm regularization in LAE (see also [10, 40, 28, 39, 16, 20, 9]). This prevents the overfitting towards the identity only to the degree that it is penalized by the $L_2$-norm regularization induced by dropout, as is outlined in the following

Let the (standard) denoising linear autoencoder (DLAE) be denoted by $\mathbf{B}^{(\text{DLAE})} := \mathbf{B}^{(\text{EDLAE})}_{a=b}$. The training objective of DLAE, as given by Eqs. 1 and 2 for $a = b$, can be simplified as follows upon convergence (i.e., in the asymptotic limit where the number of training epochs $n \to \infty$):

$$\hat{\mathbf{B}}^{(\text{DLAE})} = \underset{\mathbf{B}^{(\text{DLAE})}}{\text{argmin}} \left\| \mathbf{X} - \mathbf{X} \cdot \mathbf{B}^{(\text{DLAE})} \right\|_F^2 + \left\| \Lambda^{1/2} \cdot \mathbf{B}^{(\text{DLAE})} \right\|_F^2 \tag{3}$$

where $\mathbf{X}$ is the (finite) training data (defined above). The second term is the $L_2$-norm regularization induced by dropout-denoising, where $\Lambda^{1/2}$ is the elementwise square root of the diagonal matrix

$$\Lambda = \frac{p}{q} \cdot \text{dMat}(\text{diag}(\mathbf{X}^\top \mathbf{X})), \tag{4}$$

where $\text{dMat}(\cdot)$ denotes a diagonal matrix, $\text{diag}(\mathbf{X}^\top \mathbf{X})$ is the vector on the diagonal of $\mathbf{X}^\top \mathbf{X}$, and $p$ is the dropout-probability ($q = 1 - p$). Eq. 3 can be derived easily (e.g., by expanding the squared error into its four parts), as done in several papers (e.g., [10, 40, 28, 39, 16, 20, 9]). Eq. 3 shows that dropout-denoising has no other effect than inducing L2-norm regularization in DLAE.

### 3.1   Overfitting towards the Identity

It may seem intuitively evident that denoising by dropping out a feature in the input-layer forces the autoencoder to rely on the *other* input-features as to predict this feature in the output-layer. In this section, we outline that dropout-denoising actually does *not* prevent the autoencoder from learning the identity matrix *beyond the effect provided by the L2-norm regularization*. This can be seen most easily in the case where $\mathbf{B}^{(\text{DLAE})}_{\text{full}}$ is of full rank[2] in Eq. 3: this ridge-regression problem is solved by

$$\begin{aligned} \hat{\mathbf{B}}^{(\text{DLAE})}_{\text{full}} &= (\mathbf{X}^\top \mathbf{X} + \Lambda)^{-1} \mathbf{X}^\top \mathbf{X} = (\mathbf{X}^\top \mathbf{X} + \Lambda)^{-1} (\mathbf{X}^\top \mathbf{X} + \Lambda - \Lambda) \\ &= \mathbf{I} - (\mathbf{X}^\top \mathbf{X} + \Lambda)^{-1} \Lambda. \end{aligned} \tag{5}$$

While the identity-matrix $\mathbf{I}$ is the solution when no $L_2$-regularization is used ($\Lambda = 0$), as expected, we can see that $L_2$-regularization ($\Lambda \neq 0$) gives rise to the off-diagonal entries being different from zero in general (due to the term $(\mathbf{X}^\top \mathbf{X} + \Lambda)^{-1} \Lambda$). The off-diagonal entries are responsible for predicting feature $i$ based on the *other* features $j \neq i$. However, note that the diagonal of $\hat{\mathbf{B}}^{(\text{DLAE})}_{\text{full}}$ in Eq. 5 is still non-zero in general, and hence the value of each feature $i$ in the output-layer is *to some part* predicted by its own value in the input-layer. Hence, the overfitting towards the identity is not completely prevented by dropout-denoising (except for the futile case $p \to 1$).

## 3.2 Difference to Weight-Decay

Even though dropout-denoising does not prevent the overfitting to the identity, as just discussed, as an aside we outline in this section as to why the induced $L_2$-norm regularization nevertheless is considerably more effective in preventing overfitting than is weight decay / parameter shrinkage, which are commonly used for training deep models. In this section, we review the crucial differences and provide new insights. It can most easily be seen by considering the low-rank model $\mathbf{B}^{(\text{DLAE})} = \mathbf{U}\mathbf{V}^\top$ with one hidden layer, where $\mathbf{U}, \mathbf{V} \in \mathbb{R}^{m \times k}$ are rank-$k$ matrices. There are two key differences to weight decay:

1. In weight decay, *each* weight (model parameter) is penalized *individually*. This may be written as $||\mathbf{U}||_F^2 + ||\mathbf{V}^\top||_F^2$. In contrast, dropout induces the $L_2$-regularization $||\mathbf{B}||_F^2 = ||\mathbf{U} \cdot \mathbf{V}^\top||_F^2$. The latter regularization of the linear autoencoder $\mathbf{B}$ hence is *invariant under different parametrizations* of the same rank $k$ (e.g., whether $\mathbf{B} = \mathbf{U} \cdot \mathbf{V}^\top$, or $\mathbf{B} = \mathbf{W}_1 \cdot ... \cdot \mathbf{W}_L$ is comprised of $L$ weight-matrices in a deep model). This difference in $L_2$-regularization leads to a huge improvement in prediction accuracy in our experiments, cf. rows (1) and (2) in Table 1.

2. Instead of using a constant regularization parameter $\lambda$ across all features (like $\lambda \cdot ||\mathbf{U} \cdot \mathbf{V}^\top||_F$), dropout results in a feature-specific regularization parameter (i.e., $\Lambda$ in Eq. 4): note that it is proportional to $(\mathbf{X}^\top \mathbf{X})_{i,i}$ for input-feature $i$, which is the (uncentered) second moment of the feature's distribution. If all features were standardized (zero mean and unit variance), then this regularization became a constant $\Lambda_{i,i} = p/q$ across all features $i = 1, ..., m$. Hence, in dropout the regularization-parameter $\Lambda_{i,i}$ automatically adapts to the distribution (i.e., second moment) of each feature $i$ (see also [40, 39]). In our experiments, this makes yet another large difference, as shown in rows (2) and (3) in Table 1.

Apart from that, note that the dropout-regularization $\Lambda$ is also proportional to the ratio $p/(1-p)$ and hence increases monotonically with the dropout-probability $p$, and diverges to infinity as $p \to 1$. This section is generalized to deep networks in the Supplement.

## 4 Emphasized Denoising can prevent the Overfitting towards the Identity

EDLAE overcomes the limitations of DLAE by using two different regularizers: preventing the overfitting towards the identity is now *decoupled* from the $L_2$-norm regularization of the off-diagonal elements of $\mathbf{B}^{(\text{EDLAE})}$. The former is controlled by the weighting matrix $\mathbf{A}$ (see also Eq. 1), and the latter by the dropout-probability $p$. These facts, together with further insights outlined in the remainder of this paper, are based on the following simplification of the training-objective of Eqs. 1 and 2 for EDLAE:

**Theorem:** *In the general case $a \geq b$, the solution $\hat{\mathbf{B}}^{(\text{EDLAE})}$ determined by Eqs. 1 and 2 is identical to the solution of the following quadratic minimization problem:*

$$\hat{\mathbf{B}}^{(\text{EDLAE})} = \underset{\mathbf{B}^{(\text{EDLAE})}}{\arg\min} \left\| \mathbf{X} - \mathbf{X} \cdot \left\{ \mathbf{B}^{(\text{EDLAE})} - \text{dMat}(\text{diag}(\mathbf{B}^{(\text{EDLAE})})) \left( 1 - \frac{b}{ap + bq} \right) \right\} \right\|_F^2$$

$$+ \left\| \Lambda^{1/2} \cdot \left\{ \mathbf{B}^{(\text{EDLAE})} - \text{dMat}(\text{diag}(\mathbf{B}^{(\text{EDLAE})})) \cdot \left( 1 - \frac{\sqrt{ab}}{ap + bq} \right) \right\} \right\|_F^2 \quad (6)$$

*where $\Lambda$ is given by Eq. 4, $p$ is the dropout-probability, $q = 1 - p$, and weights $a, b$ in $\mathbf{A}$ in Eq. 1.*

**Proof:** The idea is to split the squared error in Eq. 1 into a sum over the different features, and for each feature into a sum over the different weights $a$ and $b$ in matrix $\mathbf{A}$. The five-page derivation is provided in the Supplement. □

The theorem shows that the diagonal of the matrix $\mathbf{B}^{(\text{EDLAE})}$ is *partially subtracted during training*, with the fractions $b/(ap + bq)$ and $\sqrt{ab}/(ap + bq)$ remaining in the squared-error and in the $L_2$-norm regularization, respectively. If $a > b$, we have that $\sqrt{ab}/(ap + bq) > b/(ap + bq)$, i.e., more of the diagonal of $\mathbf{B}^{(\text{EDLAE})}$ remains in the $L_2$-regularization term than in the squared-error. Based on the choices for $a$ and $b$, one can remove the diagonal to any degree. In one extreme ($a = b$), where

nothing is subtracted from the diagonal, Eq. 6 immediately simplifies to Eq. 3 for DLAE, as expected. The other extreme is $b = 0$ (while $a, p > 0$), which was named *full emphasis* in [37]: Eq. 6 above shows that the diagonal is completely subtracted from $\mathbf{B}^{(\mathrm{EDLAE})}$ during training in this case.

When the diagonal is removed to a larger degree during the training, it reduces the degree to which a feature $i$ can be reconstructed from its own value in the input-layer. Hence, this increasingly forces the model to reconstruct each feature $i$ based on all *other* features $j \neq i$ during training, and hence increasingly reduces the overfitting towards the diagonal. *In the extreme case where the diagonal is completely eliminated during training ($b = 0$), there is hence no overfitting to the diagonal.* For this reason, we focus on this case in the next section.

Another remarkable insight from the Theorem is that the diagonal plays different roles during training vs. testing/prediction: the diagonal is (partially) removed when the model is fitted to the data during training, while the learned model $\hat{\mathbf{B}}^{(\mathrm{EDLAE})}$ (i.e., with the diagonal present) is later used for making predictions (on new data points).[3]

## 4.1 Full Emphasis

When training with full emphasis ($b = 0$), there is no overfitting towards the identity, as just shown. In this section, we derive further insights for this case, for the full-rank model in Section 4.1.1, and for the low-rank model in Section 4.1.2.

### 4.1.1 Full-rank EDLAE

For full-rank[2] EDLAE, all the entries in matrix $\mathbf{B}_{\mathrm{full}}^{(\mathrm{EDLAE})}$ are independent of each other. Consequently, the diagonal values in $\mathbf{B}_{\mathrm{full}}^{(\mathrm{EDLAE})}$ are undetermined in Eq. 6 if we set $b = 0$. However, if we consider the limit $b \to 0$ for $b > 0$ and for fixed $a > 0, p > 0$, the fraction of the diagonal remaining in the squared error in Eq. 6 is proportional to $b$, while it is proportional to $\sqrt{b}$ in the $L_2$-norm regularization: on the diagonal, the regularization hence dominates over the squared error in the limit $b \to 0$ for $b > 0$–hence, we obtain for the optimal diagonal: $\mathrm{diag}(\hat{\mathbf{B}}_{\mathrm{full}}^{(\mathrm{EDLAE})}) \to 0$. Continuing with $\mathrm{diag}(\mathbf{B}_{\mathrm{full}}^{(\mathrm{EDLAE})}) = 0$, Eq. 6 now simplifies for full-rank $\mathbf{B}_{\mathrm{full}}^{(\mathrm{EDLAE})}$ with full emphasis:

$$\hat{\mathbf{B}}_{\mathrm{full}}^{(\mathrm{EDLAE})} = \underset{\mathbf{B}_{\mathrm{full}}^{(\mathrm{EDLAE})}}{\mathrm{argmin}} \left\| \mathbf{X} - \mathbf{X}\mathbf{B}_{\mathrm{full}}^{(\mathrm{EDLAE})} \right\|_F^2 + \left\| \Lambda^{1/2}\mathbf{B}_{\mathrm{full}}^{(\mathrm{EDLAE})} \right\|_F^2$$

$$\text{s.t.} \quad \mathrm{diag}(\mathbf{B}_{\mathrm{full}}^{(\mathrm{EDLAE})}) = 0 \qquad (7)$$

This optimization problem with the equality constraint $\mathrm{diag}(\mathbf{B}^{(\mathrm{EDLAE})}) = 0$ can be easily solved with the method of Lagrangian multipliers, which yields the closed-form solution

$$\hat{\mathbf{B}}_{\mathrm{full}}^{(\mathrm{EDLAE})} = \mathbf{I} - \mathbf{C} \cdot \mathrm{dMat}(\mathbf{1} \oslash \mathrm{diag}(\mathbf{C})) \qquad (8)$$

$$\text{where} \quad \mathbf{C} = \left(\mathbf{X}^\top \mathbf{X} + \Lambda\right)^{-1}, \qquad (9)$$

where $\mathbf{I}$ is the identity matrix, and $\oslash$ denotes the elementwise division by the diagonal of the matrix $\mathbf{C}$. In $\mathbf{C} \cdot \mathrm{dMat}(\mathbf{1} \oslash \mathrm{diag}(\mathbf{C}))$, each column $i$ in $\mathbf{C}$ is divided by its corresponding diagonal element $\mathbf{C}_{i,i}$. Note that the multiplication with $\mathrm{dMat}(\mathbf{1} \oslash \mathrm{diag}(\mathbf{C}))$ not only enforces the zero diagonal, but also affects the learned off-diagonal entries of $\hat{\mathbf{B}}_{\mathrm{full}}^{(\mathrm{EDLAE})}$, so that feature $i$ is best predicted by the *other* features $j \neq i$.

**Comparison of DLAE and EDLAE for full-rank models:** Even though DLAE merely applies $L_2$-norm regularization, while EDLAE with $b = 0$ completely prevents the overfitting towards the identity, the closed-form solutions of the full-rank DLAE (see Eq. 5) and the full-rank EDLAE (see Eq. 8), look surprisingly similar: the only difference is in the diagonal matrix that multiplies matrix $\mathbf{C}$. In DLAE, the diagonal of $\Lambda$ is $\frac{p}{q}\mathrm{diag}(\mathbf{X}^\top \mathbf{X})$, while in EDLAE the corresponding diagonal is $\mathbf{1} \oslash \mathrm{diag}(\mathbf{C}) = \mathbf{1} \oslash \mathrm{diag}((\mathbf{X}^\top \mathbf{X} + \frac{p}{q}\mathrm{dMat}(\mathrm{diag}(\mathbf{X}^\top \mathbf{X})))^{-1})$, which exactly enforces a zero diagonal. As the latter is an elementwise inverse of the matrix inverse, the diagonals of both models become equal in the futile limit $p \to 1$.

#### 4.1.2 Low-rank EDLAE

Even though a low-rank model obviously is unable to learn the identity exactly, it may still overfit towards it, especially in the case when the model-rank is large, for instance, about rank $k \geq 100$ in Figure 1 (left), where the prediction accuracy of the (unconstrained) low-rank model starts to be below par, even when trained with denoising.

In this section, we derive the underlying mechanism induced by low-rank EDLAE with full emphasis ($b = 0$). We consider the factorization $\mathbf{B}_{\text{low}}^{(\text{EDLAE})} = \mathbf{U}\mathbf{V}^\top$ with rank-$k$ matrices $\mathbf{U}, \mathbf{V} \in \mathbb{R}^{m \times k}$. In the low-rank model, the diagonal $\text{diag}(\mathbf{U}\mathbf{V}^\top)$ cannot be assumed to be zero in general, unlike in the full-rank model above: even though the diagonal is not explicitly determined in Eq. 6 for $b = 0$ (as it is completely eliminated), it is still indirectly determined in Eq. 6 due to the learned off-diagonal elements in $\mathbf{B}_{\text{low}}^{(\text{EDLAE})}$ due to the factorization $\mathbf{B}_{\text{low}}^{(\text{EDLAE})} = \mathbf{U}\mathbf{V}^\top$, which induces a coupling of the various entries in $\mathbf{B}_{\text{low}}^{(\text{EDLAE})}$. Eq. 6 with $b = 0$ becomes for $\mathbf{B}_{\text{low}}^{(\text{EDLAE})} = \mathbf{U}\mathbf{V}^\top$:

$$\|\mathbf{X} - \mathbf{X} \cdot \left\{\mathbf{U}\mathbf{V}^\top - \text{dMat}\left(\text{diag}(\mathbf{U}\mathbf{V}^\top)\right)\right\}\|_F^2 + \|\Lambda^{1/2} \cdot \left\{\mathbf{U}\mathbf{V}^\top - \text{dMat}\left(\text{diag}(\mathbf{U}\mathbf{V}^\top)\right)\right\}\|_F^2 \quad (10)$$

This optimization problem for $\mathbf{U}, \mathbf{V}$ can be solved efficiently with the Alternating Directions Method of Multipliers (ADMM) [13, 12, 8], see Supplement for details.

Let us now consider the interesting case where the rank $k$ of the low-rank model is 'sufficiently' large, in the sense that it can accurately approximate the full-rank solution. This is illustrated in Figure 1 (left), where the accuracy does not drop significantly for matrix-ranks as low as about $k \approx 1,000$, which is considerably smaller than the full rank of 17,769 in this experiment. In this case, one can hence expect that the diagonal of the low-rank model is approximately equal to the diagonal of the full-rank model, which is zero (see Eq. 7). For 'sufficiently' large matrix-rank $k$, the optimization problem in 10 may hence be approximated by

$$\hat{\mathbf{U}}, \hat{\mathbf{V}} = \underset{\mathbf{U}, \mathbf{V}}{\text{argmin}} \, \|\mathbf{X} - \mathbf{X}\mathbf{U}\mathbf{V}^\top\|_F^2 + \|\Lambda^{1/2}\mathbf{U}\mathbf{V}^\top\|_F^2 \qquad \text{s.t.} \quad \text{diag}(\mathbf{U}\mathbf{V}^\top) = 0 \qquad (11)$$

It can be solved with Alternating Least Squares, using closed-form updates for the optimal $\mathbf{U}$ given $\mathbf{V}$, and for the optimal $\mathbf{V}$ given $\mathbf{U}$, where the optimum can be determined analytically using the method of Lagrangian multipliers, see Supplement for details.

**Corollary: (Near-)Orthogonality Constraint:** Eq. 11 reveals that training with full emphasis causes the latent embedding $\mathbf{U}_{i,\cdot}$ in the encoder to be (approximately) *orthogonal* to the latent embedding $\mathbf{V}_{i,\cdot}$ in the decoder for each feature $i = 1, ..., m$, when using a model of 'sufficienly' large rank $k$.

*Interestingly, this (near-)orthogonality constraint is in stark contrast to the intuition that 'similar' features should have 'similar' latent embeddings.* This can be seen as follows: if two features $i$ and $j$ are similar, either one should be predictable from the other one. This means that the dot-product $\hat{\mathbf{U}}_{i,\cdot}\hat{\mathbf{V}}_{j,\cdot}^\top$ should be large. The (near-)orthogonality constraint requires, however, that $\hat{\mathbf{U}}_{j,\cdot}\hat{\mathbf{V}}_{j,\cdot}^\top = 0$ holds approximately. Hence, the latent embeddings $\hat{\mathbf{U}}_{i,\cdot}$ and $\hat{\mathbf{U}}_{j,\cdot}$ cannot be similar for similar features $i$ and $j$. The analogous argument holds for the embeddings $\hat{\mathbf{V}}_{i,\cdot}$ and $\hat{\mathbf{V}}_{j,\cdot}$ being dissimilar.

In summary, even though the (near-)orthogonality constraint requires (approximate) orthogonality of the two embeddings $\hat{\mathbf{U}}_{i,\cdot}$ and $\hat{\mathbf{V}}_{i,\cdot}$ regarding (the *same*) feature $i$ in *two different* matrices $\hat{\mathbf{U}}$ and $\hat{\mathbf{V}}$, this in turn causes the embeddings of *(*different) features $i$ and $j$ in the *same* matrix $\hat{\mathbf{U}}$ to be less similar (and analogously for the matrix $\hat{\mathbf{V}}$). This theoretical insight is also corroborated by our experiments, as illustrated in Figure 1: it shows that the cosine-similarities among 'similar' features are considerably smaller when learned with the (near-)orthogonality constraint (like in EDLAE, center graph) than without it (like in DLAE, right graph). This holds for the embeddings in the encoder $\mathbf{U}$ (in blue) and in the decoder $\mathbf{V}$ (in red) in Figure 1. In fact, the embeddings of 'similar' items are close-to-orthogonal in the encoder in the blue histogram in the center graph in Figure 1. See Section 6 for more details (like the definition of 'similar').

## 5  Related Work

The regularizer in Parseval networks [11], as well as the spread-out [42] and GLaS [14] regularizers have a similar empirical effect as the (near-)orthogonality constraint in as far as they also encourage

the learned latent vectors to become more orthogonal to each other, i.e., they also favour a small cosine-similarity, similar to the (near-)orthogonality constraint discussed in the previous paragraph. Their motivations as well as their underlying mechanisms, however, are very different from the derived (near-)orthogonality constraint, as outlined in the following. The regularizer in Parseval networks [11] is motivated by the goal of constraining the Lipschitz constant of each learned weight-layer of the model to be less than 1, as a means of improving the robustness against adversarial examples (e.g., modifications of images that are so slight that they are barely noticeable by humans, but cause the learned model to make misclassification errors with high confidence). The motivation of the spread-out [42] and GLaS [14] regularizers is to reduce the overfitting of deep networks in extreme classification problems. The regularizer in the Parseval network [11], as well as the spread-out [42] and GLaS [14] regularizers directly penalize the cosine-similarities in the *same* weight-layer of the network, i.e., they encourage all pairs of vectors *within* a layer of the network to become more orthogonal to each other. This is conceptually different from the derived (near-)orthogonality constraint, where the latent vectors *across* two layers are constrained to be orthogonal, i.e., between the two matrices $\mathbf{U}, \mathbf{V}$ (and not the vectors within $\mathbf{V}$, or within $\mathbf{U}$), see the last paragraph in the previous section for details. A second conceptual difference is that the number of constraints scales linearly with the number of features $i$ in the (near-)orthogonality constraint, while it scales quadratically in Parseval networks [11], as well as in the spread-out [42] and GLaS [14] regularizers. Third, while the derived (near-)orthogonality constraint is specific to the AE, the regularizer in the Parseval network [11] as well as the spread-out [42] and GLaS [14] regularizers can be applied to various kinds of deep networks.

Apart from that, and related to denoising / dropout, there have been several papers that viewed denoising / dropout as a form of $L_2$-norm regularization, e.g., [10, 3, 40, 28, 39, 38, 16, 20, 29, 9]: while some papers addressed aspects like speeding up computations [10, 40] or its effect on gradient descent [39, 29], a large body of work focused on a single layer [3, 39, 38, 16], or on shallow architectures [9]. Very few papers addressed denoising / dropout in deep models [20, 17]. In this paper, we considered the entire model, i.e., $\mathbf{B}$ instead of the individual layers / matrices $\mathbf{U}, \mathbf{V}$ in the model, and showed that this provides a simple yet effective explanation of the benefits (compared to weight decay) and limitations (the overfitting towards the identity is not completely prevented) of dropout-denoising. To the best of our knowledge, this paper provides the first analytic study of *emphasized denoising*.

# 6 Experiments

This section empirically illustrates the theoretical results derived in this paper, regarding the different ways of regularization. In this section, we apply autoencoders to collaborative-filtering / recommendation problems, as it focuses on the AE's ability to not overfit to the identity (see experimental set-up below). The features (columns in $\mathbf{X}$) hence correspond to the items that can be recommended, and each user corresponds to a data-point (row in $\mathbf{X}$).

**Experimental Set-Up:** For reproducibility, we follow the experimental set-up in [21], using their publicly available code as well as the same three well-known data-sets MovieLens 20 Million (*ML-20M*) [15], Netflix Prize (*Netflix*) [5], and the Million Song Data (*MSD*) [6]. Using the same pre-processing of the data as in [21], the user-item data-matrix $\mathbf{X}$ is *binary*, with 1 indicating an observed user-item interaction (e.g., user played the song). The key properties of the data-sets are summarized in Table 1. We determined the optimal training-hyper-parameters (i.e., dropout-probability $p$ and $L_2$-regularization parameter $\lambda$) using grid-search. In Table 1, the training in rows (1) and (4) was implemented in Tensorflow [33], while the models in the remaining rows were trained according to the analytic equations outlined in this paper (using Python and Numpy). The code accompanying these experiments is publicly available at `https://github.com/hasteck/EDLAE_NeurIPS2020`. As to evaluate the prediction/ranking accuracy of the learned autoencoders on the test-set, we also followed the evaluation protocol of [21] and used Normalized Discounted Cumulative Gain (nDCG@100) and Recall (@20 and @50) as in [21]. Note that the set of items $j$ in a test-user's interaction-history (which serves as input to the AE) is disjoint from the test-user's set of test-items $i$ to be predicted (see [21]). Consequently, each item $i$ has to be predicted based on the *other* items $j \neq i$, so that prediction accuracy immediately suffers in these experiments if the AE overfits to the identity function (i.e., predicts item $i$ in the output from the same item $i$ in the input).

Table 1: Prediction accuracies (Recall @20 and @50, nDCG@100, as in [21]) of a linear autoencoder of model-rank 1,000, trained with different regularizers, on three well-known data-sets: *ML-20M*, *Netflix*, and *MSD* (with standard errors 0.002, 0.001, and 0.001, respectively). Note that $\tilde{\Lambda} = \Lambda + \lambda \mathbf{I}$ has two hyper-parameters: dropout probability $p$ (see $\Lambda$ in Eq. 4) and the (scalar) $L_2$-regularization parameter $\lambda$.

| | model training | ML-20M Recall @20 | ML-20M Recall @50 | ML-20M nDCG @100 | Netflix Recall @20 | Netflix Recall @50 | Netflix nDCG @100 | MSD Recall @20 | MSD Recall @50 | MSD nDCG @100 |
|---|---|---|---|---|---|---|---|---|---|---|
| 1. | $\|\|\mathbf{X} - \mathbf{XUV}^\top\|\|_F^2 + \lambda \cdot (\|\|\mathbf{U}\|\|_F^2 + \|\|\mathbf{V}\|\|_F^2)$ | 0.345 | 0.467 | 0.376 | 0.326 | 0.406 | 0.357 | 0.200 | 0.278 | 0.249 |
| 2. | $\|\|\mathbf{X} - \mathbf{XUV}^\top\|\|_F^2 + \lambda \cdot \|\|\mathbf{U} \cdot \mathbf{V}^\top\|\|_F^2$ | 0.377 | 0.511 | 0.407 | 0.337 | 0.418 | 0.369 | 0.217 | 0.296 | 0.266 |
| 3. | $\|\|\mathbf{X} - \mathbf{XUV}^\top\|\|_F^2 + \|\|\tilde{\Lambda}^{1/2} \cdot \mathbf{U} \cdot \mathbf{V}^\top\|\|_F^2$ | 0.382 | 0.516 | 0.417 | 0.351 | 0.435 | 0.384 | 0.259 | 0.350 | 0.314 |
| 4. | DLAE: sampled dropout-denoising | 0.380 | 0.511 | 0.413 | 0.350 | 0.434 | 0.383 | 0.256 | 0.344 | 0.309 |
| 5. | EDLAE (see Section 4) | 0.389 | 0.518 | 0.420 | 0.362 | 0.446 | 0.394 | 0.263 | 0.354 | 0.320 |
| from [21] | SLIM | 0.370 | 0.495 | 0.401 | 0.347 | 0.428 | 0.379 | \-did not finish in [21]\- | | |
| | WMF | 0.360 | 0.498 | 0.386 | 0.316 | 0.404 | 0.351 | 0.211 | 0.312 | 0.257 |
| | CDAE | 0.391 | 0.523 | 0.418 | 0.343 | 0.428 | 0.376 | 0.188 | 0.283 | 0.237 |
| | MULT-VAE $^{\text{PR}}$ | 0.395 | 0.537 | 0.426 | 0.351 | 0.444 | 0.386 | 0.266 | 0.364 | 0.316 |
| | MULT-DAE | 0.387 | 0.524 | 0.419 | 0.344 | 0.438 | 0.380 | 0.266 | 0.363 | 0.313 |
| data-set properties | # items | 20,108 | | | 17,769 | | | 41,140 | | |
| | # users | 136,677 | | | 463,435 | | | 571,355 | | |
| | # interactions | 10 mil. | | | 57 mil. | | | 34 mil. | | |

**Empirical Results:** Table 1 shows the prediction-accuracies obtained for the low-rank model $\mathbf{B} = \mathbf{UV}^\top$ with matrix-rank $k = 1,000$ when trained with the various regularizations discussed in this paper. First, let us consider $L_2$-norm regularization: we can see that weight-decay (see line 1 in Table 1), which is commonly applied in training deep models, obtains by far the lowest accuracy on all three data-sets. This is greatly improved when weight-decay is replaced by the dropout-style $L_2$-regularization $\|\|\mathbf{UV}^\top\|\|_F^2$, see line 2 in Table 1. Moreover, when $\lambda$ is replaced by $\Lambda$ (see Eq. 4), which provides item-specific scaling of the L2-norm regularization, an additional large improvement is obtained, see line 3 in Table 1. Second, the results in lines 3 and 4 in Table 1 agree quite well with each other, which empirically corroborates that training with stochastic dropout is indeed asymptotically equivalent to using $L_2$-regularization in LAE, as outlined in Section 3. Third, line 5 in Table 1 shows the benefits of not overfitting towards the identity, resulting in further gains in accuracy by EDLAE over DLAE. Fourth, given that we used the same experimental protocol as in [21], our results can be compared to the various models evaluated there, including two linear models (SLIM [26], WMF [19, 27]), and three deep non-linear AEs (CDAE [41], MULT-VAE $^{\text{PR}}$ [21], MULT-DAE [21]). Table 1 shows not only that linear AEs with proper regularization can obtain competitive empirical results, even when compared to the deep non-linear baselines in Table 1, but also that the differences among the various types of regularizations can actually be substantial (i.e., possibly larger than the differences between different model-classes).

Figure 1 (left) shows that EDLAE obtains the highest prediction accuracy at *all* matrix-ranks $k$, as expected, as it completely prevents the overfitting towards the identity, while also accounting for the coupling of the diagonal and off-diagonal entries in $\mathbf{B} = \mathbf{UV}^\top$ in the low-rank model (see Section 4.1.2). We can also see that a matrix-rank of about $k \approx 1,000$ is sufficiently large (yet much smaller than the full rank $m = 17,769$) in this experiment such that training with the (near-)orthogonality constraint yields the same prediction accuracy (within standard error) as EDLAE does. This empirically corroborates our analytic derivation that the (near-)orthogonality constraint provides an accurate approximation to EDLAE for 'sufficiently' large model-ranks $k$ – which may be considerably smaller than the full rank $m$ as shown in Figure 1 (left).

In the other extreme, when the model-rank $k$ is very small, we can see that EDLAE is well approximated by applying only $L_2$-regularization (like in DLAE) in Figure 1 (left). This is not surprising given that a matrix of extremely low rank $k$ cannot approximate the identity matrix well, and hence such a low-rank model cannot overfit towards the identity. At the same time, however, prediction accuracy is quite low, due to the small small model capacity, see Figure 1 (left). As the model-rank $k$ increases and exceeds a relatively small value of about $k \approx 100$ on the *Netflix* data in Figure 1 (left), we can see that training a low-rank model with an unconstrained diagonal (like in DLAE) yields less accurate predictions than training with the (near-)orthogonality constraint. This illustrates empirically that a low-rank model may start to overfit towards the identity at already quite small model-ranks (about $k \approx 100$ in Figure 1, left), even when trained with denoising.

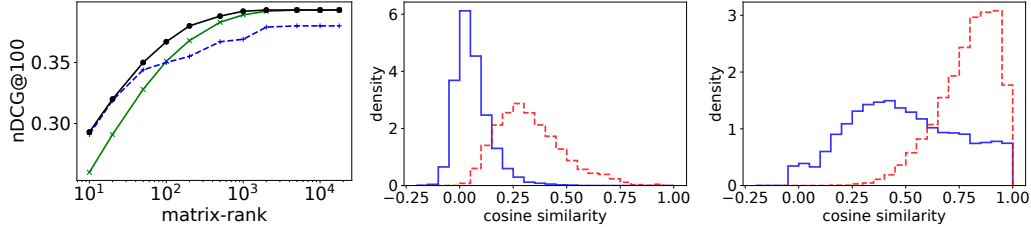

Figure 1: (Near-)orthogonality constraint (see Section 4.1.2): the **left graph** shows nDCG@100 for models of different matrix-ranks $k$, ranging from 10 to full rank. EDLAE (cf. Eq. 10; black, ●) obtains the highest accuracy by completely avoiding the overfitting towards the identity at *all* matrix-ranks $k$, as expected (see Section 4.1.2). Minimizing $||\mathbf{X} - \mathbf{X}\mathbf{U}\mathbf{V}^\top||_F^2 + \lambda||\mathbf{U}\mathbf{V}^\top||_F^2$ with the (near-)orthogonality constraint (cf. Eq. 11; green, ×), and without it (like in DLAE; blue, +) approximates EDLAE in the extreme cases of large or small matrix-ranks $k$, respectively. The **graphs in the center and on the right** show for the rank-1,000 model that the cosine-similarities of similar items in the encoder $\mathbf{U}$ (solid blue) and decoder $\mathbf{V}$ (dashed red) are considerably reduced when training with the (near-)orthogonality constraint (like in EDLAE, center graph) rather than without it (like in DLAE, right graph). All three graphs were obtained on the *Netflix* data–the results on the other two data sets were similar.

Figure 1 (center and right) shows the effect of the (near-)orthogonality constraint on the learned embeddings in the encoder $\mathbf{U}$ (in blue) and in the decoder $\mathbf{V}$ (in red): for each item $i = 1, ..., m$, we considered the 10 most similar items in the sense that they had the largest values in column $i$ of the learned full-rank (and most accurate) model $\hat{\mathbf{B}}_{\text{full}}^{(\text{EDLAE})}$ (see Eq. 8).For a given set of 10 similar items, we computed the cosine-similarities (i.e., dot product of normalized embedding-vectors) of $\mathbf{U}_i$ and $\mathbf{U}_j$ (in blue) and of $\mathbf{V}_i$ and $\mathbf{V}_j$ (in red) across all pairs $i \neq j$ in this set. Concerning each set of 10 similar items, we then took the median, and Figure 1 (center and right graph) shows the resulting distributions of these medians. As we can see, when trained with the (near-)orthogonality constraint (like in EDLAE, center graph) vs. without it (like in DLAE, right graph), the cosine-similarity of the latent embeddings of 'similar' items is vastly reduced in both the encoder and decoder. Remarkably, the embeddings of many similar items are close to orthogonal to each other in the encoder $\mathbf{U}$ (in blue in center graph), defying the intuition that similar items should have similar embeddings.

## 7 Conclusions

Considering linear autoencoders, as they facilitate analytic solutions, we first showed in this paper theoretically and empirically that denoising / dropout does not completely prevent the overfitting towards the identity-function between the input and output-layer. Instead, it merely induces $L_2$-norm regularization–which interestingly is invariant under different model-parameterizations, a key difference to weight decay that is commonly used for training deep models. To the best of our knowledge, this paper provides the first analytic study of *emphasized denosing*, which was introduced in [37]. In the main theorem derived in this paper, we show that *emphasized denoising* is essential for preventing the overfitting to the identity: interestingly, this is done by (partially) subtracting the diagonal when *learning* the model-matrix, even though the diagonal is included when *making predictions* on new data points. When subtracting the diagonal during training, the off-diagonal entries are learned such that a feature is optimally predicted from the *other* features, hence avoiding the overfitting towards the identity. When trained with full emphasis ($b = 0$), this theorem yields the closed-form solution for the full-rank EDLAE, and also reveals a new (near-)orthogonality constraint regarding the learned embeddings in the low-rank EDLAE. While conceptually different from the regularizer in Parseval networks [11] as well as the spread-out [42] and GLaS [14] regularizers, their resulting effects are empirically similar. While we limited this paper to *linear* models as to facilitate the derivation of analytic insights into the underlying mechanisms, the stochastic version of emphasized denoising is readily applicable to training deep non-linear models in practice, as done in [37], where it was empirically shown that emphasized denoising improves on (standard) denoising in deep non-linear models.

## Broader Impact

This paper provides a theoretical analysis of the overfitting problem of (linear) autoencoders to the identity-function when learned from data, and how it can be mitigated. These novel scientific insights hopefully help improve autoencoders in various practical application areas, with positive societal effects. Given that autoencoders may also be applied to collaborative filtering / recommender systems, as done in this paper for empirical illustration of the derived theoretical results, the various ethical or societal concerns that apply to collaborative filtering systems in general, like the danger of filter bubbles or various aspects of fairness, also apply here. They may be mitigated by the fact that the collaborative filtering approach (e.g., autoencoder) is typically only one component in a larger system, where several of the other components are tasked to guard against negative ethical and societal effects.

## Acknowledgements

I am very grateful to Chaitanya Ekanadham, Ashish Rastogi, and Mahdi M. Kalayeh for their valuable suggestions and comments, as well as to the anonymous reviewers for their useful feedback. I am also indebted to Dawen Liang for providing the code for the experimental setup of all three datasets. Regarding the required Financial Disclosure, I did not receive financial support beyond my employment at Netflix.

## Footnotes

[1]Note that the objective in this paper is still to minimize the reconstruction error–by predicting feature $i$ in the output-layer from the *other* features $j \neq i$ in the input-layer, i.e., without learning literally the identity function or overfitting towards it.

[2]Even though the full-rank model does not provide an encoding (like a usual AE does), it is still a useful model for making predictions. It may also be viewed as the limit of low-rank models whose rank approaches full rank. A key advantage of the full-rank model is that it allows for a closed-form solution, from which new insights can be obtained.

[3]Note that the diagonal of a low-rank $\hat{\mathbf{B}}^{(\mathrm{EDLAE})}$ does generally not vanish due to its coupling with the off-diagonal elements, see Section 4.1.2.

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
