[Supplementary Material]

# Supplement to
# 'Autoencoders that don't overfit towards the Identity'

**Harald Steck**
Netflix
Los Gatos, California
`hsteck@netflix.com`

## 1   Overview

This supplement provides

- in Section 2, the proof of the Theorem in the paper,
- in Section 3, the derivation of the ADMM equations for optimizing Eq. 10 in the paper, and
- in Section 4, the derivation of the update-equations for optimizing Eq. 11 in the paper, and
- in Section 5, the generalization of Section 3 in the paper to dropout at different layers in a deep network.

## 2 Proof of the Theorem

This first section of the proof provides an overview, where we start with the objective function of Eq. 1 in the paper (re-stated in Eq. 2 below), and show that it is equal to the objective function in the Theorem in the paper (see Eq. 8 below) up to the factor $ap + bq$, which is an irrelevant constant when optimizing for $\mathbf{B}^{(\mathrm{EDLAE})}$. But first, let us introduce the following definitions, which will be used throughout this proof:

$$
\begin{aligned}
\beta &:= \mathrm{diag}(\mathbf{B}^{(\mathrm{EDLAE})}) \\
\mathbf{B}^{(\mathrm{off})} &:= \mathbf{B}^{(\mathrm{EDLAE})} - \mathrm{dMat}(\beta)
\end{aligned}
\tag{1}
$$

where $\beta$ is the diagonal of $\mathbf{B}^{(\mathrm{EDLAE})}$, while $\mathbf{B}^{(\mathrm{off})}$ contains the off-diagonal elements of $\mathbf{B}^{(\mathrm{EDLAE})}$ and has a zero diagonal. Also note that $\mathbf{B}^{(\mathrm{EDLAE})} = q \cdot \mathbf{B}$, see Eq. 2 in the paper.

In the following, we provide the detailed steps. We first provide the sequence of manipulations at once, and then describe each step in the text below. We start by re-stating Eq. 1 in the paper

$$
\frac{1}{n} \left\| \mathbf{A}^{(n)\frac{1}{2}} \odot \left( \mathbf{X}^{(n)} - \mathbf{Z}^{(n)} \cdot \mathbf{B} \right) \right\|_F^2
\tag{2}
$$

$$
= \sum_{i=1}^{m} \left[ \frac{1}{n} \| \mathbf{A}_{\cdot,i}^{(n)\frac{1}{2}} \odot \left( \mathbf{X}_{\cdot,i}^{(n)} - \mathbf{Z}^{(n)} \cdot \mathbf{B}_{\cdot,i} \right) \|_2^2 \right]
\tag{3}
$$

$$
= \sum_{i=1}^{m} \left[ \underbrace{\frac{a}{n} \| \mathbf{X}_{\cdot,i}^{(n,i,a)} - \mathbf{Z}^{(n,i,a)} \cdot \mathbf{B}_{\cdot,i} \|_2^2}_{(a)} + \underbrace{\frac{b}{n} \| \mathbf{X}_{\cdot,i}^{(n,i,b)} - \mathbf{Z}^{(n,i,b)} \cdot \mathbf{B}_{\cdot,i} \|_2^2}_{(b)} \right]
\tag{4}
$$

$$
\xrightarrow{n \to \infty} \sum_{i=1}^{m} \left[ \underbrace{ap \left\{ \| \mathbf{X}_{\cdot,i} - \mathbf{X}\mathbf{B}_{\cdot,i}^{(\mathrm{off})} \|_2^2 + \| \Lambda^{\frac{1}{2}} \mathbf{B}_{\cdot,i}^{(\mathrm{off})} \|_2^2 \right\}}_{(a)} \right.
$$

$$
\left. + \underbrace{bq \left\{ \| \mathbf{X}_{\cdot,i} - \mathbf{X}\mathbf{B}_{\cdot,i}^{(\mathrm{off})} - \mathbf{X}\mathrm{dMat}(\frac{\beta}{q})_{\cdot,i} \|_2^2 + \| \Lambda^{\frac{1}{2}} \mathbf{B}_{\cdot,i}^{(\mathrm{off})} \|_2^2 \right\}}_{(b)} \right]
\tag{5}
$$

$$
= ap \cdot \| \mathbf{X} - \mathbf{X} \cdot \mathbf{B}^{(\mathrm{off})} \|_F^2
$$
$$
+ bq \cdot \| \mathbf{X} - \mathbf{X} \cdot \mathbf{B}^{(\mathrm{off})} - \mathbf{X} \cdot \mathrm{dMat}(\frac{\beta}{q}) \|_F^2
$$
$$
+ (ap + bq) \cdot \| \Lambda^{\frac{1}{2}} \cdot \mathbf{B}^{(\mathrm{off})} \|_F^2
\tag{6}
$$

$$
= (ap + bq) \cdot \left[ \left\| \mathbf{X} - \mathbf{X}\mathbf{B}^{(\mathrm{off})} - \mathbf{X} \cdot \mathrm{dMat}(\beta) \frac{b}{ap + bq} \right\|_F^2 \right.
$$
$$
\left. + \left\| \Lambda^{\frac{1}{2}} \mathbf{B}^{(\mathrm{off})} \right\|_F^2 + \frac{a}{b} \left\| \Lambda^{\frac{1}{2}} \beta \frac{b}{ap + bq} \right\|_2^2 \right]
\tag{7}
$$

$$
= (ap + bq) \cdot \left[ \left\| \mathbf{X} - \mathbf{X} \cdot \left\{ \mathbf{B}^{(\mathrm{EDLAE})} - \mathrm{dMat}(\mathrm{diag}(\mathbf{B}^{(\mathrm{EDLAE})})) \left( 1 - \frac{b}{ap + bq} \right) \right\} \right\|_F^2 \right.
$$
$$
\left. + \left\| \Lambda^{1/2} \cdot \left\{ \mathbf{B}^{(\mathrm{EDLAE})} - \mathrm{dMat}(\mathrm{diag}(\mathbf{B}^{(\mathrm{EDLAE})})) \cdot \left( 1 - \frac{\sqrt{ab}}{ap + bq} \right) \right\} \right\|_F^2 \right]
\tag{8}
$$

Line 2 is decomposed into a sum over the $m$ features in line 3, where $\mathbf{A}_{\cdot,i}$, $\mathbf{X}_{\cdot,i}$ and $\mathbf{B}_{\cdot,i}$ refer to column $i$.

In line 4, for each column $i$, we split column $\mathbf{A}_{\cdot,i}$ into two parts based on the two weight-values $a, b$ in $\mathbf{A}_{\cdot,i}$, and then apply the same (row-wise) split to $\mathbf{Z}^{(n)}$ (resulting in $\mathbf{Z}^{(n,i,a)}$ and $\mathbf{Z}^{(n,i,b)}$) and to $\mathbf{X}_{\cdot,i}$ (resulting in $\mathbf{X}^{(n,i,a)}$ and $\mathbf{X}^{(n,i,b)}$), see Section 2.1 for details.

Line 5 states the analytic simplifications obtained for the parts (a) and (b), respectively, when the number $n$ of training-epochs approaches infinity (for convergence). The details are outlined in Sections 2.2 and 2.3 below. See Eq. 1 above for the definitions of $\beta$ and $\mathbf{B}^{(\text{off})}$. Moreover, $\Lambda = \frac{p}{q}\text{dMat}(\text{diag}(\mathbf{X}^{\top}\mathbf{X}))$, as in Eq. 4 in the paper, is the diagonal matrix that holds the diagonal of $\mathbf{X}^{\top}\mathbf{X}$, multiplied by the dropout-probability $p$, and $q = 1 - p$.

In line 6, we change the sum over the $m$ columns back to matrix notation.

In line 7 is obtained by first expanding each of the squared terms in line 6, then collecting the corresponding parts, and finally undoing the expansions.

Finally, in line 8, we used the substitutions from Eq. 1 as to obtain $\mathbf{B}^{(\text{EDLAE})}$, and collected the corresponding terms.

## 2.1 Split into Two Parts

This section provides the details on how line 3 above is split into the two parts $(a)$ and $(b)$ as to obtain line 4 above. This is outlined in the following for a column $i \in \{1, ...m\}$. The split of the column vectors $\mathbf{A}_{\cdot,i}$ and $\mathbf{X}_{\cdot,i}^{(n)}$ as well as of the matrix $\mathbf{Z}^{(n)}$ is based on the two weights $a, b$ in the weight-vector $\mathbf{A}_{\cdot,i}$. Note that these weights reflect the fact whether the corresponding feature-value was dropped out or not, see Section 2 in the paper. To this end, we define the following two sets of row-indices:

$$
\begin{aligned}
\mathcal{R}^{(n,i,a)} &= \{\tilde{r} : \mathbf{A}_{\tilde{r},i}^{(n)} = a\} \\
\mathcal{R}^{(n,i,b)} &= \{\tilde{r} : \mathbf{A}_{\tilde{r},i}^{(n)} = b\}
\end{aligned}
$$

Given dropout-probability $p$ (and $q = 1-p$), note that the relative sizes of the sets obey $\frac{|\mathcal{R}^{(n,i,a)}|}{|\mathcal{R}^{(n,i,b)}|} \to \frac{p}{q}$ as $n \to \infty$, which will be used in the next two sections.

Based on these two sets of row-indices, we now split column $\mathbf{X}_{\cdot,i}^{(n)}$ and matrix $\mathbf{Z}^{(n)}$ as follows:

$$
\begin{aligned}
\mathbf{X}_{\cdot,i}^{(n,i,a)} &:= \mathbf{X}_{\mathcal{R}^{(n,i,a)},i}^{(n)} &\text{and}\quad \mathbf{Z}^{(n,i,a)} &:= \mathbf{Z}_{\mathcal{R}^{(n,i,a)},\cdot}^{(n)} \\
\mathbf{X}_{\cdot,i}^{(n,i,b)} &:= \mathbf{X}_{\mathcal{R}^{(n,i,b)},i}^{(n)} &\text{and}\quad \mathbf{Z}^{(n,i,b)} &:= \mathbf{Z}_{\mathcal{R}^{(n,i,b)},\cdot}^{(n)}
\end{aligned}
$$

Hence, line 3 above can be split accordingly into two parts as to obtain line 4 above:

$$
\begin{aligned}
&\frac{1}{n}||\mathbf{A}_{\cdot,i}^{(n)\frac{1}{2}} \odot \left(\mathbf{X}_{\cdot,i}^{(n)} - \mathbf{Z}^{(n)} \cdot \mathbf{B}_{\cdot,i}\right)||_2^2 \\
=\ &\underbrace{\frac{a}{n}||\mathbf{X}_{\cdot,i}^{(n,i,a)} - \mathbf{Z}^{(n,i,a)} \cdot \mathbf{B}_{\cdot,i}||_2^2}_{(a)} + \underbrace{\frac{b}{n}||\mathbf{X}_{\cdot,i}^{(n,i,b)} - \mathbf{Z}^{(n,i,b)} \cdot \mathbf{B}_{\cdot,i}||_2^2}_{(b)}
\end{aligned}
$$

## 2.2 Simplification of Part (a)

In this section, we simplify part (a) in line 4 above by eliminating $\mathbf{Z}^{(n,i,a)}$ and $\mathbf{X}_{\cdot,i}^{(n,i,a)}$, and rewriting them in terms of the given training data $\mathbf{X}$ (which is unaffected by any dropout). We first present the

equations, followed by a detailed description of the manipulations to get from one line to the next:

$$\frac{a}{n}||\mathbf{X}^{(n,i,a)}_{\cdot,i} - \mathbf{Z}^{(n,i,a)} \cdot \mathbf{B}_{\cdot,i}||^2_2 \tag{9}$$

$$= \quad \frac{a}{n}||\mathbf{X}^{(n,i,a)}_{\cdot,i} - \mathbf{Z}^{(n,i,a)}_{\cdot,-i} \cdot \mathbf{B}_{-i,i}||^2_2 \tag{10}$$

$$= \quad \frac{a}{n}(\mathbf{X}^{(n,i,a)}_{\cdot,i} - \mathbf{Z}^{(n,i,a)}_{\cdot,-i}\mathbf{B}_{-i,i})^\top \cdot (\mathbf{X}^{(n,i,a)}_{\cdot,i} - \mathbf{Z}^{(n,i,a)}_{\cdot,-i}\mathbf{B}_{-i,i}) \tag{11}$$

$$= \quad \frac{a}{n}\left\{ \underbrace{\mathbf{X}^{(n,i,a)\top}_{\cdot,i}\mathbf{X}^{(n,i,a)}_{\cdot,i}}_{1.} - \underbrace{\mathbf{X}^{(n,i,a)\top}_{\cdot,i}\mathbf{Z}^{(n,i,a)}_{\cdot,-i}}_{2.}\mathbf{B}_{-i,i} - \mathbf{B}^\top_{-i,i}\underbrace{\mathbf{Z}^{(n,i,a)\top}_{\cdot,-i}\mathbf{X}^{(n,i,a)}_{\cdot,i}}_{2.} \right.$$

$$\left. + \mathbf{B}^\top_{-i,i}\underbrace{\mathbf{Z}^{(n,i,a)\top}_{\cdot,-i}\mathbf{Z}^{(n,i,a)}_{\cdot,-i}}_{3.}\mathbf{B}_{-i,i} \right\} \tag{12}$$

$$\xrightarrow{n\to\infty} \quad ap\left\{ \underbrace{\mathbf{X}_{\cdot,i}^\top\mathbf{X}_{\cdot,i}}_{1.} - \underbrace{\mathbf{X}_{\cdot,i}^\top\mathbf{X}_{\cdot,-i}q}_{2.}\mathbf{B}_{-i,i} - \mathbf{B}^\top_{-i,i}\underbrace{q\mathbf{X}_{\cdot,-i}^\top\mathbf{X}_{\cdot,i}}_{2.} \right.$$

$$\left. + \mathbf{B}^\top_{-i,i}\underbrace{\{q^2\mathbf{X}_{\cdot,-i}^\top\mathbf{X}_{\cdot,-i} + q^2\Lambda_{-i,-i}\}}_{3.}\mathbf{B}_{-i,i} \right\} \tag{13}$$

$$= \quad ap\left\{ ||\mathbf{X}_{\cdot,i} - \mathbf{X}_{\cdot,-i}q\mathbf{B}_{-i,i}||^2_2 + ||\Lambda^{1/2}_{-i,-i}q\mathbf{B}_{-i,i}||^2_2 \right\} \tag{14}$$

$$= \quad ap\left\{ ||\mathbf{X}_{\cdot,i} - \mathbf{X}\mathbf{B}^{(\text{off})}_{\cdot,i}||^2_2 + ||\Lambda^{1/2}\mathbf{B}^{(\text{off})}_{\cdot,i}||^2_2 \right\} \tag{15}$$

When simplifying part (a), it is important that column $i$ in matrix $\mathbf{Z}^{(n,i,a)}$ is zero by construction, as all these values have been dropped out (see section above). Hence, column $i$ can be removed from matrix $\mathbf{Z}^{(n,i,a)}$ without affecting the squared loss–the resulting matrix is denoted by $\mathbf{Z}^{(n,i,a)}_{\cdot,-i}$ (where $-i$ denotes all indices except for $i$). Correspondingly, $\mathbf{B}_{-i,i}$ denotes the column-vector where row $i$ is removed from vector $\mathbf{B}_{\cdot,i}$. This yields line 10. As an aside, note that element $\mathbf{B}_{i,i}$ is not affected by part $(a)$–it is solely determined by part $(b)$, see next section.

In lines 11 and 12, the squared loss is expanded into its four terms.[1]

Line 13 is the key step, where we simplify each of the three terms as follows as the number of training-epochs $n \to \infty$:

1. As $n \to \infty$, the vector $\mathbf{X}^{(n,i,a)}_{\cdot,i}$ contains $p \cdot n \cdot r$ rows from the given data $\mathbf{X}_{\cdot,i}$ (with $r$ rows), see its precise construction in Section 2.1. Hence $\frac{1}{n}\mathbf{X}^{(n,i,a)\top}_{\cdot,i}\mathbf{X}^{(n,i,a)}_{\cdot,i} \to p\mathbf{X}_{\cdot,i}^\top\mathbf{X}_{\cdot,i}$ as $n \to \infty$, where $p$ arises from the difference in the number of rows.

2. In $\mathbf{Z}^{(n,i,a)}_{\cdot,-i}$, each entry is present with probability $q$ (see its precise construction in Section 2.1), while no entries are dropped out in vector $\mathbf{X}^{(n,i,a)}_{\cdot,i}$. We hence have $\frac{1}{n}\mathbf{X}^{(n,i,a)\top}_{\cdot,i}\mathbf{Z}^{(n,i,a)}_{\cdot,-i} \to pq\,\mathbf{X}_{\cdot,i}^\top\mathbf{X}_{\cdot,-i}$ as $n \to \infty$, where $q$ is the probability that an entry in $\mathbf{Z}^{(n,i,a)}_{\cdot,-i}$ is present, and $p$ again arises from the difference in the number of rows.

3. Given that each entry in $\mathbf{Z}^{(n,i,a)}_{\cdot,-i}$ is present with probability $q$, independently of the other entries, we hence have for all off-diagonal elements $j \neq l$ that $\frac{1}{n}\left(\mathbf{Z}^{(n,i,a)\top}_{\cdot,-i}\mathbf{Z}^{(n,i,a)}_{\cdot,-i}\right)_{j,l} \to pq^2\,(\mathbf{X}_{\cdot,-i}^\top\mathbf{X}_{\cdot,-i})_{j,l}$ as $n \to \infty$, where $q^2$ is the probability that two different entries in $\mathbf{Z}^{(n,i,a)}_{\cdot,-i}$ are both present, and $p$ again arises from the difference in the number of rows. For the diagonal elements it holds that $\frac{1}{n}(\mathbf{Z}^{(n,i,a)\top}_{\cdot,-i}\mathbf{Z}^{(n,i,a)}_{\cdot,-i})_{j,j} \to pq\,(\mathbf{X}_{\cdot,-i}^\top\mathbf{X}_{\cdot,-i})_{j,j}$, where

now $q$ (and not $q^2$) is the probability of both entries being present because both entries are the same here.

Combining the diagonal and off-diagonal entries, we obtain (as $n \to \infty$):

$$\frac{1}{n}\mathbf{Z}_{\cdot,-i}^{(n,i,a)\top}\mathbf{Z}_{\cdot,-i}^{(n,i,a)} \quad \to \quad p \cdot \left\{ q^2\mathbf{X}_{\cdot,-i}^{\top}\mathbf{X}_{\cdot,-i} + (q-q^2)\mathrm{dMat}(\mathrm{diag}(\mathbf{X}_{\cdot,-i}^{\top}\mathbf{X}_{\cdot,-i})) \right\}$$

$$= \quad p \cdot \left\{ q^2\mathbf{X}_{\cdot,-i}^{\top}\mathbf{X}_{\cdot,-i} + q^2\Lambda_{-i,-i} \right\} \tag{16}$$

where we used $q - q^2 = q^2\frac{p}{q}$, and defined $\Lambda_{-i,-i} = \frac{p}{q}\mathrm{dMat}(\mathrm{diag}(\mathbf{X}_{\cdot,-i}^{\top}\mathbf{X}_{\cdot,-i}))$, where $\Lambda_{-i,-i}$ means that column $i$ and row $i$ are removed from matrix $\Lambda = \frac{p}{q}\cdot\mathrm{dMat}(\mathrm{diag}(\mathbf{X}^{\top}\mathbf{X}))$.

In line 14, we collect the terms (reversing the earlier expansion) and obtain the squared loss plus the remainder, which is an L2-norm regularization term (where the square root is applied elementwise to $\Lambda$).

Finally, line 15 is obtained by using the identities $\mathbf{B}_{-i,i}^{(\mathrm{off})} = \mathbf{B}_{-i,i}^{(\mathrm{EDLAE})}$ for the off-diagonal elements $j \neq i$ (see Eq. 1 above), and $\mathbf{B}_{-i,i}^{(\mathrm{EDLAE})} = q\mathbf{B}_{-i,i}$ (see Eq. 2 in the paper). Moreover, given that element $\mathbf{B}_{i,i}^{(\mathrm{off})} = 0$, we can undo the removal of element $i$ from vector $\mathbf{B}^{(\mathrm{off})}$, and correspondingly from matrices $\mathbf{X}$ and $\Lambda$, as it does not change the squared norms.

## 2.3 Simplification of Part (b)

In this section, we simplify part $(b)$ in line 4 above, by eliminating $\mathbf{Z}^{(n,i,b)}$ and $\mathbf{X}_{\cdot,i}^{(n,i,b)}$, and rewriting them in terms of the given training data $\mathbf{X}$ (which is unaffected by dropout). Like above, we first present the equations, followed by a detailed description of the manipulations to get from one line to the next:

$$\frac{b}{n}||\mathbf{X}_{\cdot,i}^{(n,i,b)} - \mathbf{Z}^{(n,i,b)} \cdot \mathbf{B}_{\cdot,i}||_2^2 \tag{17}$$

$$= \quad \frac{b}{n}||\mathbf{X}_{\cdot,i}^{(n,i,b)} - \mathbf{Z}^{(n,i,b)} \cdot \underbrace{\mathbf{Q}\mathbf{Q}^{-1}}_{=\mathbf{I}}\mathbf{B}_{\cdot,i}||_2^2 \tag{18}$$

$$= \quad \frac{b}{n}\left\{ \underbrace{\mathbf{X}_{\cdot,i}^{(n,i,b)\top}\mathbf{X}_{\cdot,i}^{(n,i,b)}}_{1.} - \underbrace{\mathbf{X}_{\cdot,i}^{(n,i,b)\top}\mathbf{Z}^{(n,i,b)}\mathbf{Q}}_{2.}\mathbf{Q}^{-1}\mathbf{B}_{\cdot,i} - \mathbf{B}_{\cdot,i}^{\top}\mathbf{Q}^{-1}\underbrace{\mathbf{Q}\mathbf{Z}^{(n,i,b)\top}\mathbf{X}_{\cdot,i}^{(n,i,b)}}_{2.} \right.$$

$$\left. + \mathbf{B}_{\cdot,i}^{\top}\mathbf{Q}^{-1}\underbrace{\mathbf{Q}\mathbf{Z}^{(n,i,b)\top}\mathbf{Z}^{(n,i,b)}\mathbf{Q}}_{3.}\mathbf{Q}^{-1}\mathbf{B}_{\cdot,i} \right\} \tag{19}$$

$$\xrightarrow{n\to\infty} \quad bq\left\{ \underbrace{\mathbf{X}_{\cdot,i}^{\top}\mathbf{X}_{\cdot,i}}_{1.} - \underbrace{\mathbf{X}_{\cdot,i}^{\top}\mathbf{X}q}_{2.}\mathbf{Q}^{-1}\mathbf{B}_{\cdot,i} - \mathbf{B}_{\cdot,i}^{\top}\mathbf{Q}^{-1}\underbrace{q\mathbf{X}^{\top}\mathbf{X}_{\cdot,i}}_{2.} \right.$$

$$\left. +\mathbf{B}_{\cdot,i}^{\top}\mathbf{Q}^{-1}\underbrace{\{q^2\mathbf{X}^{\top}\mathbf{X} + q^2[\Lambda]_{(i)}\}}_{3.}\mathbf{Q}^{-1}\mathbf{B}_{\cdot,i} \right\} \tag{20}$$

$$= \quad bq\left\{ ||\mathbf{X}_{\cdot,i} - \mathbf{X}\mathbf{Q}^{-1}q\mathbf{B}_{\cdot,i}||_2^2 + \mathbf{B}_{\cdot,i}^{\top}\mathbf{Q}^{-1}q^2[\Lambda]_{(i)}\mathbf{Q}^{-1}\mathbf{B}_{\cdot,i} \right\} \tag{21}$$

$$= \quad bq\left\{ ||\mathbf{X}_{\cdot,i} - \mathbf{X}\mathbf{Q}^{-1}q\mathbf{B}_{\cdot,i}||_2^2 + ||[\Lambda^{1/2}]_{(i)}\mathbf{Q}^{-1}q\mathbf{B}_{\cdot,i}||_2^2 \right\} \tag{22}$$

$$= \quad bq\left\{ ||\mathbf{X}_{\cdot,i} - \mathbf{X}\mathbf{Q}^{-1}\mathbf{B}_{\cdot,i}^{(\mathrm{EDLAE})}||_2^2 + ||[\Lambda^{1/2}]_{(i)}\mathbf{Q}^{-1}\mathbf{B}_{\cdot,i}^{(\mathrm{EDLAE})}||_2^2 \right\} \tag{23}$$

$$= \quad bq\left\{ ||\mathbf{X}_{\cdot,i} - \mathbf{X}\mathbf{Q}^{-1}\mathbf{B}_{\cdot,i}^{(\mathrm{EDLAE})}||_2^2 + ||\Lambda^{1/2}\mathbf{B}_{\cdot,i}^{(\mathrm{off})}||_2^2 \right\} \tag{24}$$

$$= \quad bq\left\{ ||\mathbf{X}_{\cdot,i} - \mathbf{X}\mathbf{B}_{\cdot,i}^{(\mathrm{off})} - \mathbf{X}\cdot\mathrm{dMat}(\frac{\beta}{q})_{\cdot,i}||_2^2 + ||\Lambda^{1/2}\mathbf{B}_{\cdot,i}^{(\mathrm{off})}||_2^2 \right\} \tag{25}$$

We start in line 17 by re-stating part $(b)$ from line 4 above. In line 18 we introduce the diagonal matrix $\mathbf{Q}^{(i)}$ (and $\mathbf{I}$ denotes the identity matrix) such that $\mathbf{Q}^{(i)}_{jj} = 1$ for $j \neq i$, and $\mathbf{Q}^{(i)}_{ii} = q$. The reason for introducing $\mathbf{Q}^{(i)}$ is that matrix $\mathbf{Z}^{(n,i,b)}$ was constructed in Section 2.1 such that it contains exactly those rows where none of the entries in column $i$ were dropped out. In all other columns $j \neq i$ of $\mathbf{Z}^{(n,i,b)}$, the entries were dropped out independently of each other with probability $q = 1 - p$. For this reason, we now *scale* the values in column $i$ by the factor $q$, which is the purpose of matrix $\mathbf{Q}^{(i)}$. In the equations above, we simplified the notation $\mathbf{Q}^{(i)}$ to $\mathbf{Q}$ for easier readability.

In lines 19, we expanded the squared error into its four terms, like in part $(a)$ before.

Line 20 is the key step, where we simplify the the three terms as follows as $n \to \infty$:

1. As $n \to \infty$, the vector $\mathbf{X}^{(n,i,b)}_{\cdot,i}$ contains $q \cdot n \cdot r$ rows from the given data $\mathbf{X}_{\cdot,i}$ (with $r$ rows), see its precise construction in Section 2.1. Analogous to the same term in part (a), we have $\frac{1}{n}\mathbf{X}^{(n,i,b)\top}_{\cdot,i}\mathbf{X}^{(n,i,b)}_{\cdot,i} \to q\mathbf{X}^\top_{\cdot,i}\mathbf{X}_{\cdot,i}$ as $n \to \infty$, where $q$ arises from the difference in the number of rows.

2. In $\mathbf{Z}^{(n,i,b)}\mathbf{Q}^{(i)}$, the entries are present with probability $q$ in all columns $j \neq i$, while column $i$ is uncorrupted but scaled by $q$ due to $\mathbf{Q}^{(i)}$. Hence, given the (uncorrupted) vector $\mathbf{X}^{(n,i,b)}_{\cdot,i}$, we have $\frac{1}{n}\mathbf{X}^{(n,i,b)\top}_{\cdot,i}\mathbf{Z}^{(n,i,b)}\mathbf{Q}^{(i)} \to q^2\mathbf{X}^\top_{\cdot,i}\mathbf{X}$ as $n \to \infty$, where one $q$ arises from the difference in the number of rows, and the other $q$ is due to the dropout/scaling with $q$.

3. Again, we have to consider the diagonal and off-diagonal entries separately, and further have two cases on the diagonal:

   - off-diagonal entries: $\left(\frac{1}{n}\mathbf{Q}^{(i)}\mathbf{Z}^{(n,i,b)\top}\mathbf{Z}^{(n,i,b)}\mathbf{Q}^{(i)}\right)_{j,l} \to q^3(\mathbf{X}^\top\mathbf{X})_{j,l}$ as $n \to \infty$ for all $j \neq l$. Again, one $q$ arises from the difference in the number of rows, while the remaining $q^2$ is the probability that two different entries in $\mathbf{Z}^{(n,i,b)}$ were both present after the dropout for $j, l \neq i$; in the case that $j = i$ or $l = i$, one was present with probability $q$ and one was scaled with $q$ (by $\mathbf{Q}^{(i)}$).

   - diagonal entry $(i,i)$: given that column $i$ was scaled with $q$ (by $\mathbf{Q}^{(i)}$), we get $\left(\frac{1}{n}\mathbf{Q}^{(i)}\mathbf{Z}^{(n,i,b)\top}\mathbf{Z}^{(n,i,b)}\mathbf{Q}^{(i)}\right)_{i,i} \to q^3(\mathbf{X}^\top\mathbf{X})_{i,i}$ as $n \to \infty$, where again one $q$ arises from the difference in the number of rows, while the remaining $q^2$ is due to the scaling of column $i$ with $q$ (by $\mathbf{Q}^{(i)}$).

   - diagonal entries $(j,j)$ for $j \neq i$: given that entries are present with probability $q$ in column $j \neq i$, we get $\left(\frac{1}{n}\mathbf{Q}^{(i)}\mathbf{Z}^{(n,i,b)\top}\mathbf{Z}^{(n,i,b)}\mathbf{Q}^{(i)}\right)_{j,j} \to q^2(\mathbf{X}^\top\mathbf{X})_{j,j}$ as $n \to \infty$, where again one $q$ arises from the difference in the number of rows, while the other $q$ is due to the probability $q$ of each entry being present.

   Combining the diagonal and off-diagonal entries, we obtain (as $n \to \infty$):

   $$\frac{1}{n}\mathbf{Q}^{(i)}\mathbf{Z}^{(n,i,b)\top}\mathbf{Z}^{(n,i,b)}\mathbf{Q}^{(i)} \quad \to \quad q \cdot \left\{q^2\mathbf{X}^\top\mathbf{X} + (q - q^2)\mathrm{dMat}([\mathrm{diag}(\mathbf{X}^\top\mathbf{X})]_{(i)})\right\}$$
   $$= \quad q \cdot \left\{q^2\mathbf{X}^\top\mathbf{X} + q^2[\Lambda]_{(i)})\right\}$$

   where $[\cdot]_{(i)}$ denotes that entry $i$ in the vector (or entry $(i,i)$ in the diagonal matrix) is set to 0; the second line is obtained by again using $q - q^2 = q^2\frac{p}{q}$, and $\Lambda = \frac{p}{q}\mathrm{dMat}(\mathrm{diag}(\mathbf{X}^\top\mathbf{X}))$.

In line 21, we collect the terms as to obtain the squared error, plus the remainder, which is an L2-norm regularization term. The latter term can also be written in terms of a Frobenius norm, which yields line 22.

Line 23 is obtained by using the identity $\mathbf{B}^{(\mathrm{EDLAE})} = q\mathbf{B}$, see Eq. 2 in the paper.

In line 24, we simplify the $L_2$-norm regularization term as follows: because $[\Lambda]_{(i)}$ is zero for entry $(i,i)$, we can first drop $\mathbf{Q}^{(i)-1}$, as its diagonal is different from 1 only for index $i$, and then replace $\mathbf{B}^{(\mathrm{EDLAE})}_{\cdot,i}$ with $\mathbf{B}^{(\mathrm{off})}_{\cdot,i}$ (see also Eq. 1 above), given that here its entry $(i,i)$ gets multiplied by 0 from $[\Lambda]_{(i)}$. Given that now entry $\mathbf{B}^{(\mathrm{off})}_{i,i} = 0$, we can replace $[\Lambda]_{(i)}$ by $\Lambda$ in the L2 regularization. In other words, we 'moved' the zero entry in $[\Lambda]_{(i)}$ to $\mathbf{B}^{(\mathrm{off})}_{\cdot,i}$ without changing the value of the $L_2$-norm regularization term.

In line 25, we rewrite the diagonal matrix $\mathbf{Q}^{(i)-1}$ as follows: we first use the identity $\mathbf{B}^{(\mathrm{EDLAE})} = \mathbf{B}^{(\mathrm{off})} + \mathrm{dMat}(\beta)$, where $\beta = \mathrm{diag}(\mathbf{B}^{(\mathrm{EDLAE})})$, as defined in Eq. 1 above. Now we see that in $\mathbf{B}_{\cdot,i}^{(\mathrm{EDLAE})}$, only the entry $\mathbf{B}_{i,i}^{(\mathrm{EDLAE})} = \beta_{i,i}$ gets affected by $\mathbf{Q}^{(i)-1}$, as the diagonal of $\mathbf{Q}^{(i)-1}$ is different from one only at $i$, where it is $1/q$.

## 3  ADMM-updates to minimize Eq. 10 in the Paper

We first re-write Eq. 10 in the paper in the following equivalent form:

$$\|\mathbf{X} + \mathbf{X} \cdot \mathrm{dMat}(\beta) - \mathbf{X}\mathbf{U}\mathbf{V}^\top\|_F^2 + \|\Lambda^{1/2}\mathbf{U}\mathbf{V}^\top\|_F^2 - \|\Lambda^{1/2}\beta\|_2^2$$
$$\text{s.t.} \quad \mathrm{diag}(\mathbf{U}\mathbf{V}^\top) = \beta \tag{26}$$

as to decouple the training-updates for the matrices $\mathbf{U}$ and $\mathbf{V}$ from the diagonal $\beta := \mathrm{diag}(\mathbf{U}\mathbf{V}^\top)$. This constrained least-squares problem can efficiently be solved using the Alternating Directions Method of Multipliers (ADMM) [3, 2, 1] as follows: the equality constraint $\mathrm{diag}(\mathbf{U}\mathbf{V}^\top) = \beta$ is absorbed into the augmented Lagrangian

$$\|\mathbf{X} + \mathbf{X} \cdot \mathrm{dMat}(\beta) - \mathbf{X}\mathbf{U}\mathbf{V}^\top\|_F^2 + \|\Lambda^{1/2}\mathbf{U}\mathbf{V}^\top\|_F^2 - \|\Lambda^{1/2}\beta\|_2^2$$
$$+ 2\gamma^\top\Omega(\beta - \mathrm{diag}(\mathbf{U}\mathbf{V}^\top)) + \|\Omega^{1/2}(\beta - \mathrm{diag}(\mathbf{U}\mathbf{V}^\top))\|_2^2 \tag{27}$$

where $\gamma$ is the vector of Lagrangian multipliers that (upon convergence of ADMM) enforces the equality constraint $\mathrm{diag}(\mathbf{U}\mathbf{V}^\top) = \beta$. Instead of a scalar penalty parameter (as used in the review [1]), we here use a vector of penalty parameters, one for each feature; $\Omega$ is a diagonal matrix, and $\mathrm{diag}(\Omega)$ is this vector of penalty parameters. $\Lambda$ and $\Omega$ are training-hyper-parameters, while $\mathbf{U}$, $\mathbf{V}$, $\beta$, and the vector of Lagrangian parameters $\gamma$ are learned. Given any two of $\mathbf{U}, \mathbf{V}, \beta$, the third one can be optimized in closed form by setting the derivative of the augmented Lagrangian to zero, and then solving for it. As a result, we obtain the following update-equations of the iterative ADMM algorithm (see also [1]):

$$\hat{\mathbf{U}} \leftarrow (\mathbf{X}^\top\mathbf{X} + \Lambda + \Omega)^{-1}\left(\mathbf{X}^\top\mathbf{X} \cdot \mathrm{dMat}(\mathbf{1} + \hat{\beta}) + \Omega \cdot \mathrm{dMat}(\hat{\beta} - \hat{\gamma})\right)\hat{\mathbf{V}}(\hat{\mathbf{V}}^\top\hat{\mathbf{V}})^{-1}$$

$$\hat{\mathbf{V}}^\top \leftarrow \left(\hat{\mathbf{U}}^\top(\mathbf{X}^\top\mathbf{X} + \Lambda + \Omega)\hat{\mathbf{U}}\right)^{-1}\hat{\mathbf{U}}^\top\left(\mathbf{X}^\top\mathbf{X} \cdot \mathrm{dMat}(\mathbf{1} + \hat{\beta}) + \Omega \cdot \mathrm{dMat}(\hat{\beta} - \hat{\gamma})\right)$$

$$\hat{\beta} \leftarrow \frac{\mathrm{diag}(\mathbf{X}^\top\mathbf{X}\hat{\mathbf{U}}\hat{\mathbf{V}}^\top) - \mathrm{diag}(\mathbf{X}^\top\mathbf{X}) + \mathrm{diag}(\Omega) \odot (\mathrm{diag}(\hat{\mathbf{U}}\hat{\mathbf{V}}^\top) + \hat{\gamma})}{\mathrm{diag}(\mathbf{X}^\top\mathbf{X}) + \mathrm{diag}(\Omega - \Lambda)}$$

$$\hat{\gamma} \leftarrow \hat{\gamma} + \mathrm{diag}(\hat{\mathbf{U}}\hat{\mathbf{V}}^\top) - \hat{\beta}$$

where the last line updates the vector of Lagrangian multipliers, see [1]. In the update of the vector $\beta$, the fraction denotes an elementwise division of the vectors, and $\odot$ is the elementwise multiplication. As to ensure that this division is well-defined, we chose $\Omega = \Lambda + \omega\mathbf{I}$, where $\omega > 0$ is a scalar, so that we are left with a single penalty parameter (rather than the vector $\mathrm{diag}(\Omega)$) that needs to be tuned in the training.

## 4  Optimization of Eq. 11 in the Paper

Using the method of Lagrangian multipliers, the two matrices $\mathbf{U}$ and $\mathbf{V}$ can be learned by minimizing Eq. 11 in the paper as follows. The equality constraint $\mathrm{diag}(\mathbf{U}\mathbf{V}^\top) = 0$ is absorbed in the Lagrangian

$$\|\mathbf{X} - \mathbf{X}\mathbf{U}\mathbf{V}^\top\|_F^2 + \|\Lambda^{\frac{1}{2}}\mathbf{U}\mathbf{V}^\top\|_F^2 + 2\eta^\top \cdot \mathrm{diag}(\mathbf{U}\mathbf{V}^\top) \tag{28}$$

where $\eta$ is the vector of Lagrangian multipliers.

For fixed $\hat{\mathbf{U}}$, the optimal $\hat{\mathbf{V}}^\top$ can be found in closed form by setting the derivative of the Lagrangian to zero, and solving for $\hat{\mathbf{V}}^\top$:

$$\hat{\mathbf{V}}^\top = \underbrace{\left(\hat{\mathbf{U}}^\top(\mathbf{X}^\top\mathbf{X} + \Lambda)\hat{\mathbf{U}}\right)^{-1}}_{=:\mathbf{D}}\hat{\mathbf{U}}^\top\left(\mathbf{X}^\top\mathbf{X} - \mathrm{dMat}(\eta)\right) \tag{29}$$

where the vector of Lagrangian multipliers $\eta$ is determined by the constraint $\text{diag}(\mathbf{U}\mathbf{V}^\top) = 0$, resulting in

$$\eta = \frac{\text{diag}(\mathbf{U}\mathbf{D}\mathbf{U}^\top\mathbf{X}^\top\mathbf{X})}{\text{diag}(\mathbf{U}\mathbf{D}\mathbf{U}^\top)} \tag{30}$$

where the division of the two diagonals is elementwise.

Conversely, for fixed $\hat{\mathbf{V}}$, Eq. 28 is minimized by

$$\hat{\mathbf{U}} = \mathbf{C}\left(\mathbf{X}^\top\mathbf{X} - \text{dMat}(\eta)\right)\hat{\mathbf{V}}(\hat{\mathbf{V}}^\top\hat{\mathbf{V}})^{-1} \tag{31}$$

where $\mathbf{C} = (\mathbf{X}^\top\mathbf{X} + \Lambda)^{-1}$. Note that matrix $\hat{\mathbf{V}}^\top\hat{\mathbf{V}}$ is invertible, as it is of rank $k$ and of size $k \times k$ (see above), and hence full rank. The (column-)vector of Lagrangian multipliers $\eta$ is again determined by the constraint $\text{diag}(\mathbf{U}\mathbf{V}^\top) = 0$, which yields to the following system of linear equations

$$\text{diag}(\mathbf{C}\mathbf{X}^\top\mathbf{X}\hat{\mathbf{V}}(\hat{\mathbf{V}}^\top\hat{\mathbf{V}})^{-1}\hat{\mathbf{V}}^\top) = \left(\mathbf{C} \odot (\hat{\mathbf{V}}(\hat{\mathbf{V}}^\top\hat{\mathbf{V}})^{-1}\hat{\mathbf{V}}^\top)\right) \cdot \eta \tag{32}$$

which has to be solved for $\eta$. Here, $\odot$ denotes the elementwise product of the two $m \times m$ matrices. Note that $\eta$ in Eqs. 30 and 32 converge towards each other, as they refer to the same vector of Lagrangian multipliers in Eq. 28.

Like in Alternating Least Squares, these two updates of $\hat{\mathbf{U}}$ and $\hat{\mathbf{V}}$ are iterated until convergence (which took a few dozen iterations in our experiments).

## 5   $L_2$-norm Regularization in Deep Networks

As an aside regarding Section 3 in the paper (concerning denoising and dropout), here we briefly discuss dropout at different layers in a deep feed-forward network. Again, we consider linear models for analytic tractability. While a deep *linear* network is not more expressive than a shallow one, its study may still help to better understand deep nonlinear models, e.g. [4].

Let us consider a network with $L + 1$ layers and the weight matrices $\mathbf{W}_1, ..., \mathbf{W}_L$. This results in $\mathbf{B} = \mathbf{W}_1 \cdot ... \cdot \mathbf{W}_L$ (as a generalization of $\mathbf{B} = \mathbf{U}\mathbf{V}^\top$ in Section 3.2 in the paper). If we apply dropout to layer $j \in \{1, ..., L\}$, the derivation in the paper (Eqs. 3 and 4) immediately carries over: the induced $L_2$-norm regularization applies to the product of the weight matrices that *follow* the dropout-layer $j$: $||\Lambda^{(j)1/2} \cdot \mathbf{W}_j \cdot ... \cdot \mathbf{W}_L||_F^2$. The $L_2$-regularization is scaled by the diagonal matrix $\Lambda^{(j)}$, which depends on the second moment of the distribution of each node in the dropout-layer $j$, in other words $\text{diag}(\Lambda^{(j)}) = \frac{p}{q}\text{diag}(\mathbf{X}^{(j)\top}\mathbf{X}^{(j)})$ where $\mathbf{X}^{(j)} = \mathbf{X} \cdot \mathbf{W}_1 \cdot ... \cdot \mathbf{W}_{j-1}$ are the (predicted) values of the hidden nodes in layer $j$.

This also reveals that, if dropout is applied to *several* different layers of a network, each dropout-layer induces the corresponding L2-regularization. Consequently, weight matrices that are positioned closer to the output-layer are subject to a larger number of $L_2$-regularization terms than weight matrices that are located closer to the input-layer of the network.

## Footnotes

[1]In our notation, the indices take priority over transposition, e.g., $\mathbf{B}^\top_{-i,i} = (\mathbf{B}_{-i,i})^\top$.