[Reviews · NeurIPS 2020]

Review 1

Summary and Contributions: The authors provide a theoretical analysis of emphasized denoising linear autoencoders (EDLAE). After showing that using dropout in denoising auto encoders (DAE) induces L2 regularization, they extend the theoretical analysis to EDLAE and show that, contrary to DAE, EDLAE prevents an autoencoder to learn an identity function.

Strengths: * The paper is nicely written and not too hard to follow for someone who's not deep in the field.

Weaknesses: * L20: The premise of this work is that an AE that learns the identity mapping between input and output is a futile solution. Perhaps I am missing the point (in which case the authors should explain better), but an autoencoder that learns the perfect identity mapping is not necessarily useless: it is possible that the latent space is meaningful, and yet the AE perfectly reconstructs its inputs. I assume the authors mean identity function not as in "perfect reconstruction", but that the AE is the actual identity function (meaning that each feature is autoencoded independently to the others)? Even if this may be obvious from the context, it would be more clear to make this distinction.

Correctness: * The claims seem correct.

Clarity: * It is largely easy to follow the main thread through the paper, and equations are properly explained.

Relation to Prior Work: * The authors provide a good basis of prior work. * The related work is not explained in enough detail, most papers are grouped and explained in half a sentence.

Reproducibility: Yes

Additional Feedback: * Figure 1 would benefit from having a legend. * L53: typo, wer -> were * Eq. 6: typo, closing bracket missing for B^{(EDLAE)} under the argmin +++++++++++++++++++ Final comments after rebuttal After reading the rebuttal and discussing with the other reviewers, I recommend accept. +++++++++++++++++++


Review 2

Summary and Contributions: This paper tackles the issue that AEs may overfit to identity function. It theoretically analyze the linear AE and show that denosing/dropout AEs only prevents overfitting to a limited extend while emphasized denoising AE can completely eliminate the overfitting issue.

Strengths: - Desnoisng autoencoders and dropout are commonly adopted techniques to prevent overfitting in AE models. This paper theoretically analyzed this type of AE and show that denoising only indces L2-norm regularization. Even with such techniques, the model is still at the risk of overfitting towards the identity function. - The authors theoretically proves that the emphasized denosing AE is able to completely avoid the overfitting issue. They also unveil the connection between emphasized denoising AE with spread-out and GLaS regularizers.

Weaknesses: - This paper only states that the AE model may still overfit towards identity even when low-rank model is used in Section 4.1.2. But I don't see any justification about this statement. Is this just a property of a naive AE model or is this also a property shared by the denoising AE model? If a low-rank denosing AE model can prevent this problem, then in practice the denoising AE will not overfit to identity since the latent dimension is much lower than the input dimension. - Both the theoretical analysis and the experiments are conducted on linear AE. It is quite common to analyze a simplified model for convenience. But to see how the conclusions drawn from simpler cases can help us understand more complex cases, I think the experiments using more complex models can be added to make the arguments stronger.

Correctness: Yes, the claims and the methods in this paper are correct.

Clarity: This paper is well-written and easy to follow.

Relation to Prior Work: The relations to prior works are clearly discussed.

Reproducibility: Yes

Additional Feedback: ========= After author feedback ======== I have read all the reviews and the author feedback. My score remains unchanged.


Review 3

Summary and Contributions: This submission considers linear regularized autoencoders, for which the authors reproduce the closed form solution, showing it to tend towards the identity as the regularization parameter vanishes. They then design a family of alternative loss functions/criteria EDLAE defined over parameters (a,b) which relax current criteria; for a certain regime of (a,b), the EDLAE criterion collapses to DLAE, while for other regimes this "overemphasizes" a property of the criterion. The authors provide an ADMM derivation to optimize the matrix system. They demonstrate results on three recommendation problems.

Strengths: This paper is very thorough analytically, and provides a good overview of implicit priors/biases of L2 regularization or, equivalently, dropout denoising. The authors have moved to a simplified case (in comparison to e.g. deep auto-encoders) so that the method in question in analytically tractable. This allows for the derivation of a central theorem providing a family of solution criteria, which relax "traditional" DLAE losses. The resulting set of solution matrices express features as linear combinations of other features _without_ self-referrence (or, for non-extremal hyper parameter values, for varying levels of self-referrence), but only during training. In another approximation the auto-encoder is forced to have diagonal = 0, which means that for a test datapoint x^*, each output index of x^*B is a combination of the other indices. Continuing the authors intuition, this forces a distributed representation of each of the x^* indices. This last fact/intuition is, in my opinion, understated. This method appears to work very well on a recommender task, modulo my experimental concerns outlined below, competitive with a 2018 WWW deep method (from which the data and experiments were taken).

Weaknesses: This paper is necessarily limited in scope, in order to preserve analytic tractability. While it's clear that the authors are aware of this, it limits the relevant audience of the paper. However, I think this paper _should_ have much broader appeal, but focusing on the analytic necessities early on prevents this. I have outlined below some specific problems, but overall I think that the paper should make a stronger point of explaining why overfitting to the identity is a problem; the algebra explains already that it does happen. If the reader is unconcerned with the applications, then they can ignore that text, but if they don't understand why they would want non-identity auto-encoders then such an explanation, even short, would be beneficial. The experimental setup is not well described. The metric in Table 1 is _not_ prediction accuracy, but nDCG@100, which the authors conflate (there isn't a single notion of accuracy in recommender systems, as the use of nDCG@100 suggests). Further, the task at hand is obfuscated and unclear, and this is a major disservice to the previous sections: the experiments are on a recommendation task, where each user has overly sparse descriptors. They may have interacted with/watched/liked only a small subset of the possible items/features (marked as 1), but when presented with any of the "negatively marked" (i.e. 0) features they may still interact/watch them. Thus, the data conflate negatives with non-reporting, and the task is to recover a ranking of the "negatives" which, when actually reported without conflation, will result in the highest number of interactions. This is measured empirically by holding out some interactions in each user descriptor vector. I understand that the paper is very full, but the only reason I understood these experiments is due to the single sentence "We follow the experimental setup in [19]", and then subsequently reading citation [19]. This setup is not at all described in this paper, which is unfortunate because it renders the experiments much less meaningful. I strongly recommend adding text which puts the these experiments in context. Further, the cited paper is a deep learning method, and these results seem comparibly strong! This is a very good thing*, but this fact is obfuscated by this missing information. The missing metrics from [19] can also be included a supplement, though of course this is optional. *caveat: I cannot tell if the experiments are actually comparable. I suspect that they are, but it would make the paper much stronger if it could be said definitively. I also struggled with the intuition of "overfitting to the identity" for some time. After reading sections 3 and 4 I eventually understood the value of "not fitting to the identity", but this could have come at a much earlier juncture. Specifically, in the introduction the paragraph "In denoising, when a feature is randomly dropped out from a data-point in the input layer, then the (uncorrupted) value of this feature in the output layer has to be predicted based on the other features of that data-point. It hence seems intuitively evident that this may prevent the AE from overfitting towards the identity. The fact that this intuition turns out to be only partially correct, as we will show, motivated this work." hints at this intuition. For the naive reader such as myself, please make the intuition painfully apparent. Something similar to "Fitting an identity matrix is not useful. We fit auto-encoders to learn about interactions, dependencies, and statistical structures between features; fitting an identity matrix is exactly not helpful for that learning objective. Thus, it is important to find solutions that are disparate from the identity." (nitpick) Dropout and L2 regularization are equivalent asymptotically, but this is in the large sample large batch limit. It's perhaps unfair to say that it "merely induces L2", though the point is clear.

Correctness: Yes.

Clarity: With respect to prose, yes.

Relation to Prior Work: Yes.

Reproducibility: Yes

Additional Feedback: (optional suggestion) For readability I would restate Eq.1 in "the Theorem". The Broader Impact statement and the broader impacts of e.g. recommender systems are within the scope of the paper, contrary to the included statement. Please reconsider the submitted statement. ==== Updated Review after Rebuttal/Discussion: Thank you for submitting additional information and answering questions as space allowed. I think my comments about the motivation were misunderstood; it becomes apparent in the manuscript why we want to avoid fitting an identity function, but this only happens after a thorough reading. I think it would benefit readers if this happened immediately, perhaps using text similar to the rebuttal paragraph. Not all autoencoding tasks have this same intuition or context, so narrowing the apparent scope at the start may be helpful. The experimental results shown in the rebuttal table and explanation are very helpful, and I hope it is the intention of the authors to include it in the final paper. The broader impacts statement need only read "This work does not present any foreseeable societal consequence.", if the authors believe that to be the case. Helpful references: https://nips.cc/Conferences/2020/PaperInformation/NeurIPS-FAQ http://www.brenthecht.com/papers/FCADIscussions_NegativeImpactsPost_032918.pdf


Review 4

Summary and Contributions: The paper discusses the problem of preventing an autoencoders from overfitting to the identity mapping, focusing on linear autoencoders for greater tractability. It exploits the idea of "emphasised denoising" and derives analytical solutions in the limit of infinite training set. Experiments on benchmark datasets show statistically significant improvement over several other regularisation methods.

Strengths: Strong theoretical paper. Thorough discussion of results. The proofs are explained in detail.

Weaknesses: Not much empirical evaluation (could have added more in the Appendix). It is not clear from the paper how much the algorithm described is useful in practical applications.

Correctness: As far as I was able to judge, yes.

Clarity: Overall, yes. Some things were not clear to me: is the random Kronecker delta matrix used to construct the weighting matrix A^(n) the same as the one used to construct Z? There is a typo in line 158 ("trainiung"). Lines 154-156 are not clear to me. What is it precisely that the Authors consider "remarkable"? The diagonal is penalised, so it's not surprising that it will be partially removed. The proofs in the Supplementary Material should restate the theorem, since there are no space limitations there. It will make it easier to follow. I would recommend using ISO-4 abbreviations for journal names in References.

Relation to Prior Work: Mostly yes. A reference for Alternating Least Squares method is missing.

Reproducibility: Yes

Additional Feedback: It would be interesting to discuss how emphasized denoising depends on the correlation between features, in particular how this correlation affects the optimal hyperparameter values. === AFTER AUTHOR FEEDBACK === Most of my questions and comments were not addressed in the Feedback (I assume because of space constraints), hence the score remains unchanged. Unlike one of the other reviewers, I agree with the substance of the Broader Impact statement. The paper doesn't deal with recommender systems in particular. I don't think it would be possible to write anything meaningful about its broader impact via this particular application of autoencoders. I would, however, recommend that the Authors rethink the style and tone of the Broader Impact statement. I think a more formal way of expressing it would less likely to inspire controversy and divert attention from their scientific contribution.


Review 5

Summary and Contributions: Since auto-encoders aim to reconstruct the input at the output, a common failure mode is overfitting towards the identity mapping. To avoid this, various types of regularization schemes are employed, one of which involves denoising (in particular, the case of dropout is considered in the paper). In the spirit of the various papers (starting with the work of Wager et al, Wang et al.) studying dropout in feedforward networks theoretically, and uncovering the regularization effects afforded by it, this paper studies the effect of denoising autoencoders for avoiding overfitting towards the identity. To enable simplicity of analysis only linear autoencoders are considered. The contributions of the paper are two-fold: 1) It shows that denoising auto-encoders don't fully prevent overfitting to identity. They do so only to the extent afforded by a suitable L2 penalty. 1.5) The effects of dropout (and the corresponding L2 penalty) are better than other penalties such as weight decay etc. for the same problem. 2) It then proposes a variant dubbed as emphasized denoising autoencoder which is able to get beyond the issue discussed. To expand on the above points: - The setting considered is when the AE is represented by matrix B, which is assumed to be low rank. B decomposes as U V^T, where U serves as the encoder and V the decoder. The usual setup using dropout is encapsulated in equation (1) for training and equation (2) for testing. Section 3 of the paper explicates on point (1) outlined above. That is, denoising prevents overfitting to the identity only to the extent that it is penalized by a suitable L2 penalty. The objective is given in equation (3) and this has obvious similarity with much of the older works on dropout in linear neural networks as discussed in the deep learning book by Goodfellow et al and the works of Wager et al. and Sida Wang et al. The fact that denoising does not prevent overfitting to the identity can be seen clearly through equation (5): the prediction of each feature in the output to some extent is predicted by its own value in the input. - To prevent overfitting towards the identity, the authors consider equation (5) and decouple the influence of the off diagonal elements leading to two separate terms as shown in equation (6). The intuition to why this prevents overfitting to the identity is clear: removal of the diagonal term to a much larger degree prevents reconstruction of some feature i at the output from its own value at the input, and is forced to depend on the other features. How much of the diagonal is removed is what is referred to as "emphasis", with full emphasis being b=0 when all of the diagonal is removed. - Section 4 carries the analysis based on the above forward. First for a full rank model, which culminates in equation (8) which looks eerily similar to equation (5), except that the former enforces a zero diagonal. The analysis is then repeated for a low rank model (followed by a low rank model but with sufficiently large rank) which is registered in equation (10). The interesting observation about this setting is that that full emphasis based training induces near-orthogonality among the latent embeddings. The optimization can be carried out via ADMM. The above observations are corroborated in recommendation systems based tasks (MovieLens, Netflix and MSD). Comparisons are made to the various scenarios discussed above and an improvement is shown in the emphasized denoising AE .

Strengths: The paper is well written and has a clean framework. It uncovers a somewhat non-obvious facet of denoising auto-encoders: that they are not fully immune to overfitting towards learning the identity mapping. The paper demonstrates this using standard arguments (standard in the sense that they are what one might follow for the analysis of dropout) and proposes a regularization that alleviates this problem. The framework also gives a justification for the success of near-orthogonality constraints on latent embeddings used in various different works for regularization. The paper also gives a theoretical justification for the work referenced in [33].

Weaknesses: The paper's weakness follows from its strengths. I think the paper presents a very clean analysis of an issue with denoising autoencoders. However, the arguments used are pretty standard at this point. While I do think that the contributions are interesting in any case, the paper would have far great value if the experimental analysis makes up by being more thorough (for example like in [13]). It would be useful to have baselines (just for the sake of comparison) so that the reader gets an idea of what numbers are good (although I do understand that this is not the point of the paper). It would also be useful to report results for other measures as used in [19] and not just nDCG@100). More importantly, it would be great if the authors can show how the insights that they have obtained for linear encoders can be transferred to training non-linear autoencoders experimentally and how prevention of over-fitting can help obtain good embeddings. As some of the references in the paper indicate, this is a major research question.

Correctness: The claims and the empirical methodology are correct.

Clarity: The paper is very well written. It helps that the framework is very clean (owing to the fact that only linear autoencoders are considered). Minor comments: Page 2, line 53: "linear models wer studied" -- typo Page 4, line 158: "When trainiung" typo in Page 5 line 165: "Applying Occam’s razor, we continue with the simplest choice, namely" I think it is reasonable to mention that only the simplest choice is taken.

Relation to Prior Work: Discussion of prior work is reasonably thorough.

Reproducibility: No

Additional Feedback: In my opinion, the broader impact section could be better worded. If the authors so surmise that there are no direct ethical considerations, it is fair game to just say so directly.

[Author Response · NeurIPS 2020]

We would like to thank all five (!) reviewers for their detailed reviews and their suggestions / questions, which will help
to further improve this paper. In the following we will try to address the main points raised.

**Experiments (reviewers 2, 3, 4, 5):** Given that our main contribution is the theoretical analysis of (emphasized)
denoising as a training technique, and its ability/lack of preventing the autoencoder (AE) from overfitting to the identity-
function, the space remaining for the experimental section is naturally limited. We nevertheless aim for experimental
reproducibility as well as an empirical comparison to other baselines by exactly following the experimental set-up in
[19]. Based on the reviews, we will make our paper more self-contained, and add a short review of the experimental
protocol in [19]. In the table below, we also added the various models evaluated in [19] for ease of comparison: two
linear models (SLIM , WMF), and three deep non-linear AEs (CDAE, MULT-VAE $^{PR}$, MULT-DAE)–we will also add their
citations to the paper. All approaches in this table can be compared to each other: this shows not only that EDLAE
(linear model) obtains competitive results compared to the various (non-linear) baselines, but also that the differences
among the various types of regularizations can actually be substantial (i.e., possibly larger than the differences between
different model-classes). In the table below, we also added Recall @20 and @50, the two metrics we had omitted in
the paper, as they largely reflect the same behavior as nDCG@100 does (in more detail, the table shows that EDLAE
empirically improves in particular the ranking accuracy in the top-$N$ for *smaller $N$*). While we limited this paper to
*linear* models for reasons of analytical tractability (see paper for the various derived insights), in practice the stochastic
version of emphasized denoising is readily applicable to training deep non-linear models, as done in [33], where it was
shown that emphasized denoising empirically improves on (standard) denoising.

**Identity Function (reviewers 1, 3):** We will clarify the motivation/objective at the beginning of this paper in more
detail. Due to space constraints, we had unfortunately shortened this part of the paper too much, as we now realize.
There are many applications where the data may be noisy or where we want the AE to be able to generalize to unseen
data (e.g., in the areas of image processing, information retrieval, etc.). Learning the identity function (i.e., predicting
each feature $i$ in the output layer from the *same* feature $i$ in the input layer) is obviously not useful for such prediction
tasks. Instead, the AE has to learn all the relevant dependences/interactions among the features, as to achieve maximum
prediction accuracy on unseen noisy test-data. Intuitively speaking, when the learned AE makes predictions for a
feature $i$ in the output-layer by relying 'too much' on *the same* feature $i$ in the input layer (i.e., identity function), and
'not enough' on the *other* features it depends on, we call this 'overfitting towards the identity function' in this paper.
In fact we chose collaborative filtering on implicit feedback data for our experiments exactly because the value 0 in
the user-item training-matrix conflates true negative items (which the user would never select) and the true positive
items that the user has not selected yet in the observed (training-)data: predicting the positives in the disjoint test-set
hence hinges on the AE's ability to predict each feature/item $i$ from the *other* items $j \neq i$, i.e., prediction accuracy
immediately suffers in our experiments if the AE overfits to the identity function.

**Low-rank models & Denoising (reviewer 2):** While fully emphasized denoising (controlled by parameters $a > b = 0$)
completely eliminates the 'overfitting toward the identity function', i.e., diagonal of matrix $\mathbf{B}$ (see Section 4 in the
paper), note that this is *decoupled* from the amount of L2-norm regularization applied to the off-diagonal entries of
$\mathbf{B}$ (which is controlled by the value of dropout-probability $p$, or $\Lambda$), see Eq. 6. In contrast, this decoupling is absent
(1) when using (standard) dropout-denoising, which merely induces L2-norm regularization in a linear model (in the
asymptotic limit, i.e., when trained to convergence, even on a finite amount of training data), and hence regularizes
both the diagonal and off-diagonal entries in the same way (see also Eq. 5); (2) when using low-rank models, where a
decrease in the model-rank not only reduces the overfitting towards the identity function, but also the model-capacity in
general, possibly leading to under-fitting for small model-ranks. Due to this coupling, the overfitting to the identity can
only be prevented partially without suffering from under-fitting the data when using only low-rank and/or denoising,
resulting in worse ranking-metrics in the table below (cf. rows 1-4 vs. EDLAE).

**We find it remarkable in l. 154-6 (reviewer 4)** that training (diagonal removed) differs from prediction (with diagonal).

|  |  | *ML-20M* | | | *Netflix* | | | *MSD* | | |
|---|---|---|---|---|---|---|---|---|---|---|
|  | model training: | Recall @20 | Recall @50 | nDCG @100 | Recall @20 | Recall @50 | nDCG @100 | Recall @20 | Recall @50 | nDCG @100 |
| 1. | $\|\|\mathbf{X} - \mathbf{XUV}^\top\|\|_F^2 + \lambda \cdot (\|\|\mathbf{U}\|\|_F^2 + \|\|\mathbf{V}\|\|_F^2)$ | 0.345 | 0.467 | 0.376 | 0.326 | 0.406 | 0.357 | 0.200 | 0.278 | 0.249 |
| 2. | $\|\|\mathbf{X} - \mathbf{XUV}^\top\|\|_F^2 + \lambda \cdot \|\|\mathbf{U} \cdot \mathbf{V}^\top\|\|_F^2$ | 0.376 | 0.508 | 0.407 | 0.342 | 0.423 | 0.374 | 0.222 | 0.303 | 0.270 |
| 3. | $\|\|\mathbf{X} - \mathbf{XUV}^\top\|\|_F^2 + \|\|\tilde{\Lambda}^{1/2} \cdot \mathbf{U} \cdot \mathbf{V}^\top\|\|_F^2$ | 0.382 | 0.515 | 0.417 | 0.351 | 0.434 | 0.384 | 0.258 | 0.347 | 0.311 |
| 4. | DLAE (sampled) | 0.383 | 0.515 | 0.417 | 0.351 | 0.435 | 0.384 | 0.257 | 0.346 | 0.311 |
| 5. | EDLAE | 0.389 | 0.518 | 0.420 | 0.359 | 0.443 | 0.392 | 0.263 | 0.354 | 0.320 |
| from [19] | SLIM | 0.370 | 0.495 | 0.401 | 0.347 | 0.428 | 0.379 | –did not finish in [19]– | | |
|  | WMF | 0.360 | 0.498 | 0.386 | 0.316 | 0.404 | 0.351 | 0.211 | 0.312 | 0.257 |
|  | CDAE | 0.391 | 0.523 | 0.418 | 0.343 | 0.428 | 0.376 | 0.188 | 0.283 | 0.237 |
|  | MULT-VAE $^{PR}$ | 0.395 | 0.537 | 0.426 | 0.351 | 0.444 | 0.386 | 0.266 | 0.364 | 0.316 |
|  | MULT-DAE | 0.387 | 0.524 | 0.419 | 0.344 | 0.438 | 0.380 | 0.266 | 0.363 | 0.313 |

[Meta-Review · NeurIPS 2020]

All reviwers have appreciated the theoretical analysis in the paper that highlights that denoising autoencoder can still overfit towards identity. The paper also proposes a way to avoid this. Authors should address the points raised in the reviews about more clarity on experimental setting, giving more intuition on why its important to prevent overfitting towards identity, and commenting on how the intuitions can be carried forward to non-linear autoencoders. Authors should also consider rephrasing the BI statement to something more formal as pointed out in one of the reviews.